# Correlated-photon time- and frequency-resolved optical spectroscopy

**Raúl Álvarez-Mendoza** [1,5], **Lorenzo Uboldi** [2,5], **Ashley Lyons** [1],
**Richard J. Cogdell** [3], **Giulio Cerullo** [2,4] ✉ **& Daniele Faccio** [1] ✉

Classical time-resolved optical spectroscopy experiments are performed using sequences of ultrashort light pulses, with photon fluxes incident on the sample which are many orders of magnitude higher than real-world conditions corresponding to sunlight illumination. Spectroscopy and microscopy schemes that use quantum states of light have been widely described theoretically with fewer experimental demonstrations that typically require very long measurements that can extend for hours or more. Here, we show that time-resolved spectroscopy with quantum light can be performed without compromising measurement time or wavelength tunability, recording a fluorescence lifetime trace in biological samples in less than a second with acceptable signal-to-noise ratio. Starting from spontaneous parametric down-conversion driven by a continuous-wave laser, we exploit the temporal correlation between randomly generated signal/idler pairs to obtain temporal resolution, and their spectral correlation to select the excitation frequency. We also add spectral resolution in detection, using a 'photon-efficient' Fourier transform approach which employs a common-path interferometer. We demonstrate the potential of this approach by resolving, at the single-photon level, excitation energy transfer cascades from LH2 to LH1 in the photosynthetic membrane and disentangling the lifetimes of two dyes in a mixture. Our results provide a new approach to ultrafast optical spectroscopy, where experiments are performed under illumination intensity conditions comparable to real-world sunlight illumination.

Optical spectroscopy is a powerful investigation tool for the microscopic mechanisms underlying various physical, chemical and biological processes. Since the development of mode-locked laser systems, ultrafast optical spectroscopy has played a pivotal role in unveiling the early snapshots of photophysical events with picosecond to femtosecond time resolution[1]. The studied phenomena, relevant both for a fundamental physico-chemical understanding and for technological applications, include, for example, charge photogeneration in photovoltaic materials[2], rate-limiting steps in photosynthesis[3], the mechanism of human vision[4] and non-equilibrium lattice dynamics in solids[5].

In standard implementations of ultrafast spectroscopy, the system under study is subjected to a series of light pulses whose frequencies and time delays are the control knobs of the experiment[6]. The simplest experiments are performed by exciting samples with an ultrashort light pulse and detecting their time-dependent fluorescence or absorption changes. Such experiments are described using a semiclassical formulation of light-matter interaction in which light pulses are treated as electromagnetic waves, neglecting their quantum nature. Recently, a growing body of theoretical studies[7–10] has proposed the exploitation of quantum states of light to enhance ultrafast optical spectroscopy. Most of

[1]School of Physics and Astronomy, University of Glasgow, Glasgow, UK. [2]Department of Physics, Politecnico di Milano, P.zza L. da Vinci 32, Milano, Italy. [3]School of Molecular Biosciences, University of Glasgow, Glasgow, UK. [4]CNR-Institute for Photonics and Nanotechnologies (CNR-IFN), Milan, Italy. [5]These authors contributed equally: Raúl Álvarez-Mendoza, Lorenzo Uboldi. ✉e-mail: giulio.cerullo@polimi.it; daniele.faccio@glasgow.ac.uk

these schemes take advantage of quantum correlations, using squeezed states of light or entangled photon pairs (EPPs) generated by spontaneous parametric down-conversion (SPDC). Quantum light offers several advantages for spectroscopy[11], such as increasing signal-to-noise ratio, providing novel control knobs (such as energy, time and polarization entanglement of the EPPs) for designing experiments or even generating completely new signals with respect to classical light. Examples of theoretical proposals of the use of quantum light for time-resolved spectroscopy are controlling exciton pathways in molecular aggregates by absorption of EPPs[12], use of EPPs to probe exciton-exciton interactions[13,14], control of azobenzene photoisomerization yield via the entanglement time of EPPs[15] and fluorescence-detected two-dimensional spectroscopy using EPPs to increase spectral resolution and reveal cross-peaks[16]. Pathway selectivity[17], time-energy resolution[18], molecular selectivity[19] and two-dimensional spectroscopy[20,21] can also be improved in a broader context using quantum light. Theoretical studies have also shown that, in principle, careful choice and shaping of classical probe states can reproduce some aspects of quantum measurements performed with classical pump pulses and single-photon Fock-state probe states[22].

Despite the plethora of theoretical proposals, experimental demonstrations of spectroscopy with quantum light are, to date, very limited. Notable examples are the linear scaling with the light intensity of the two-photon absorption rate for time-frequency EPPs[23], the use of quantum correlations to enhance nonlinear optical microscopy/spectroscopy, pushing the sensitivity of stimulated Raman[24] or Brillouin[25,26] scattering or time-domain THz spectroscopy[27] below the shot-noise limit, and the use of squeezed states of light to enhance sensitivity and generate extra nonlinear signals in four-wave-mixing of rubidium vapors[28].

Recently, a pioneering study by Li et al.[29] used time-resolved photon-counting quantum light spectroscopy to study excitation energy transfer (EET) processes in the light-harvesting 2 (LH2) complex of the purple photosynthetic bacterium *Rhodobacter sphaeroides*, containing two rings of bacteriochlorophylls (BChls), B800 and B850. In that study, an EPP at 808 nm was generated by SPDC from a pulsed ultrashort laser. One of the two photons served as a heralding photon and was detected by a single-photon avalanche diode (SPAD) while the other was used to excite the B800 BChls in the LH2. Following EET to the B850 BChls, which occurs on the picosecond timescale[30], the fluorescence photon emitted by the LH2 was detected by a second SPAD and its delay with respect to the heralding photon was measured by an event timer, thus reconstructing the fluorescence lifetime by time-correlated single photon counting (TCSPC). By correlating the heralding and the detected photons, this experiment demonstrated that photosynthetic EET from B800 to B850 proceeds at the single-photon level.

In this work we exploit the correlations that result from the entanglement between photon pairs to demonstrate frequency and sub-200-ps time-resolved fluorescence following single-photon excitation from a continuous wave (CW) laser. Whilst traditional time-resolved measurements are performed using ultrafast pulsed laser systems, here we demonstrate that by exploiting the frequency and time correlation properties of EPPs, it is possible to perform high signal-to-noise ratio (SNR) measurements in short acquisition times using a CW laser. We extend the experiment by Li et al. in two important ways. First, we generate the EPP starting from a CW laser and exploit the temporal correlations between the photons generated by the SPDC process to achieve ~200-ps temporal resolution and their spectral correlations to achieve resolution in excitation frequency. We then resolve the spectrum of the fluorescence photons using a time-domain Fourier transform approach, employing an ultra-stable birefringent interferometer[31]. We thus exploit the photon correlations that arise from entanglement to measure time-resolved fluorescence

spectra following single-photon excitation from a CW laser and, by combining temporal and spectral resolution, achieve a "correlated photon streak camera". We demonstrate the potential of this spectroscopic tool in two experiments: we resolve, at the single-photon level, EET cascades within a photosynthetic membrane and disentangle the lifetimes of two dyes in a mixture. Critically, we highlight the broad impact and applicability of our approach beyond a proof-of-principle demonstration, by recording TCSPC time traces of light-harvesting complexes with acquisition times down to sub-second.

## Experimental setup

Figure 1a shows a schematic overview of the approach and a more detailed overview of the experimental setup is shown in Fig. 1b–d. The signal photon from an SPDC photon-pair excites a sample, generating a fluorescent photon that is detected by a SPAD sensor. This detection however, is heralded by the idler photon both spectrally (by placing an interference filter in front of the herald SPAD) and temporally (through the TCSPC electronics). The experiment is implemented starting with a CW single-frequency blue laser ($\lambda = 413$ nm) focused in a 30-mm-long periodically poled KTiOPO$_4$ (ppKTP) crystal designed for type 0 quasi phase matching, with poling period $\Lambda = 3.675$ μm. Typical laser power is 0.25 mW and the beam is collimated to a diameter of 400 μm, corresponding to a confocal parameter of 600 mm. At a temperature of 56 °C, ppKTP generates EPPs at 800 and 860 nm, which are separated by a dichroic mirror (DM in Fig. 1a). The 860-nm photon, which acts as heralding photon, is selected by an interference bandpass filter (BP in Fig. 1a) and is coupled via a single-mode fiber (SMF in Fig. 1a) to a SPAD. The 800-nm photon is focused via a polarizing beam splitter and a 0.65-numerical aperture microscope objective to a cuvette containing the molecular sample to be investigated. The sample fluorescence is collected in the back-scattering direction via the same microscope objective and then coupled via a graded index multi-mode fiber (MMF in Fig. 1a) to a second SPAD. The two SPAD outputs are sent to a TCSPC unit which monitors the delay of the fluorescence photon with respect to the heralding photon and, by building a histogram of these delays, measures the fluorescence lifetime. We underline that the measurements shown here with a CW pump laser exploit the simultaneous correlations in time and frequency that are a direct consequence of the entanglement of the photons at the point of generation in the nonlinear crystal. Moreover, differently from TCSPC systems which use a pulsed laser for photoexcitation[32], here the excitation laser is CW and the time resolution is provided by the temporal correlations between the heralding and the signal photon. That said, we also underline that this approach does not explicitly rely on entanglement to achieve any form of 'quantum advantage' in the strictest sense (i.e. a measurement that could not be achieved by any conceivable classical means).

To add spectral resolution to the system we use Fourier transform detection, by placing in front of the second SPAD a birefringent interferometer, the Translating-Wedge-based Identical pulses eNcoding System (TWINS, Fig. 1d, see "Methods" for a detailed description)[31], which creates two delayed replicas of the incident optical waveform with interferometric delay stability. By recording a series of TCSPC traces for different replicas delays and performing a Fourier transform, one obtains a time-resolved fluorescence spectrum following single-photon excitation. Importantly, the use of TWINS implies that the fluorescence photon can be efficiently collected on a simple bucket SPAD detector rather than with a SPAD array (or a scanning SPAD detector) that would instead be required with a grating spectrometer, leading to higher losses, lower signal-to-noise ratio (the photons are spread across multiple SPADs) and more complicated detection hardware. We note that the TWINS spectrometer requires polarized light at the input and that the polarizing beamsplitter used to collect the fluorescence leads to 50% loss.

The choice of a Fourier transform spectrometer approach is based on SNR considerations alone. With the TWINS, we collect all the light

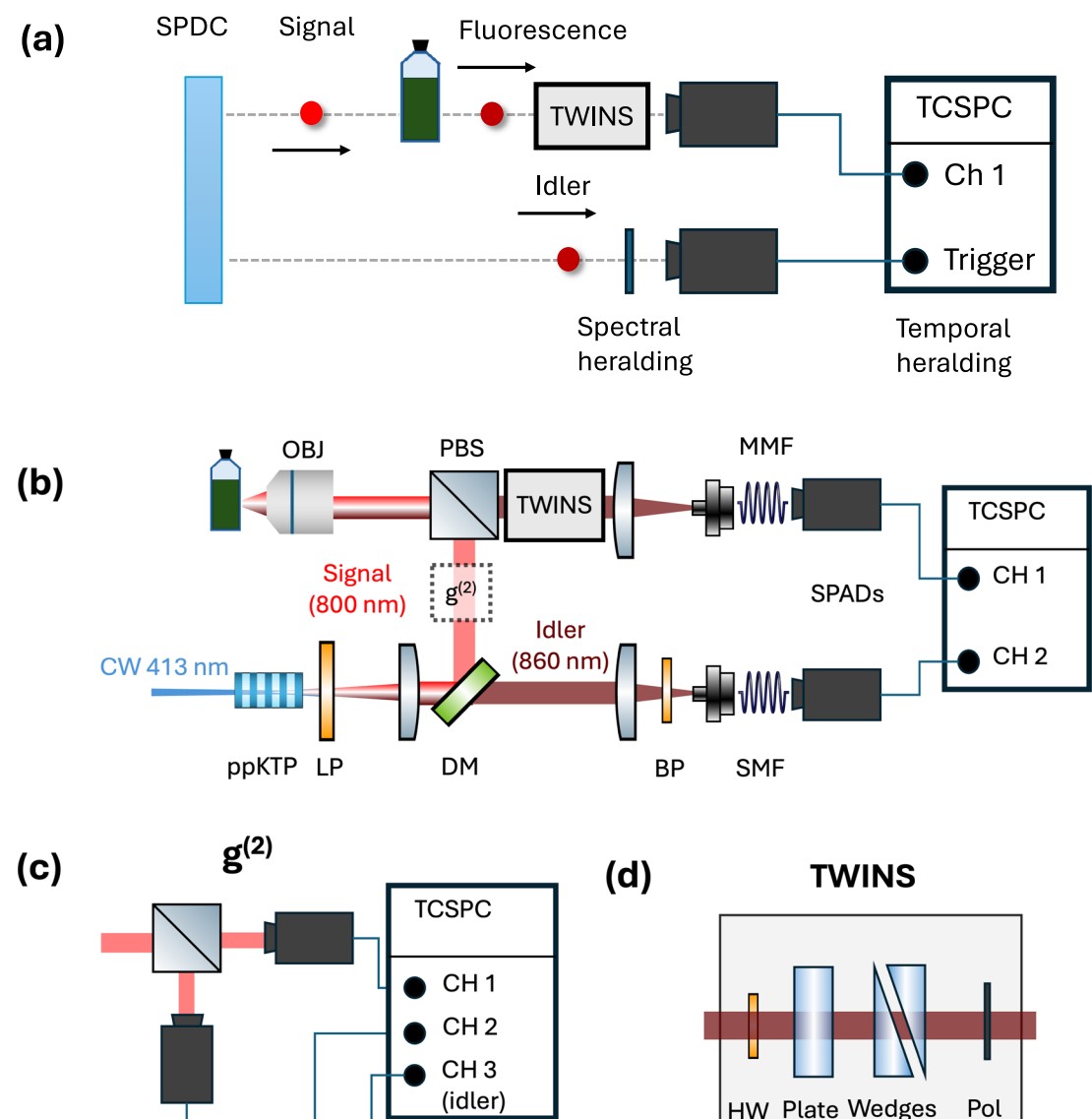

**Fig. 1 | Time- and frequency-resolved spectroscopy with correlated photons.**
**a** Schematic overview of spectral (interference filter on the idler arm) and temporal heralding of fluorescent photons. `TWINS' indicates the Fourier Transform spectrometer used to spectrally resolve the fluorescence. **b** Experimental setup. LP: long-pass filter; DM: dichroic mirror; PBS: polarizing beam splitter used to select a single input polarization for the TWINS spectrometer; BP: bandpass filter; OBJ: microscope objective; SMF: single-mode fiber; MMF: multi-mode fiber; SPAD: single-photon avalanche diode; TCSPC: time-correlated single photon counting. **c** Setup for $g^{(2)}$ measurement. **d** Scheme of the TWINS interferometer. HW: half-waveplate; Plate: $\alpha$-BBO plate; Wedges: movable $\alpha$-BBO wedges; Pol: polarizer.

with a single SPAD that is optimized for photon collection (100%) and quantum efficiency (70% at 800 nm). This ensures that high SNR time traces at each interferometer position can be acquired for short times with high signal and low dark counts. The compromise is that acquisition times are of course lengthened by the requirement to scan over interferometer delays in our experiments. A different approach could be to use a grating spectrometer coupled e.g. to a SPAD array. In the latter case, quantum efficiencies are typically a factor 10× lower and fill factors can typically also be a factor 10× lower compared to a single pixel SPAD. Moreover, the spectrum is spatially dispersed resulting in an N-fold signal reduction, where N is the number of pixels or spectral points measured. This signal can be recovered by increasing the acquisition time so as to match the total TWINS acquisition time (that is lengthened due to the M scan points). We can estimate the ratio, R, of the SNR for the single-pixel TWINS and grating spectrometer coupled to a SPAD array as (see "Methods")

$$R = \frac{\sqrt{S + D_{\mathrm{camera}} \cdot m}}{\sqrt{M}\sqrt{S + D_{\mathrm{pixel}}}} \tag{1}$$

where $S \sim 100$ counts is the measured signal in 1 s, $D$ is the sensor dark counts (for the camera and single pixel), $m = N/q$ and $q$ is the photon detection probability reduction in the camera compared to a single pixel. Inserting values from our experiments for the TWINS case and using published values for state-of-the-art SPAD cameras (see Methods), we find that for the same total acquisition time, $R \sim 4$–8, indicating that there is an advantage in using a Fourier Transform approach over the competing solution using a grating spectrometer coupled to a SPAD camera.

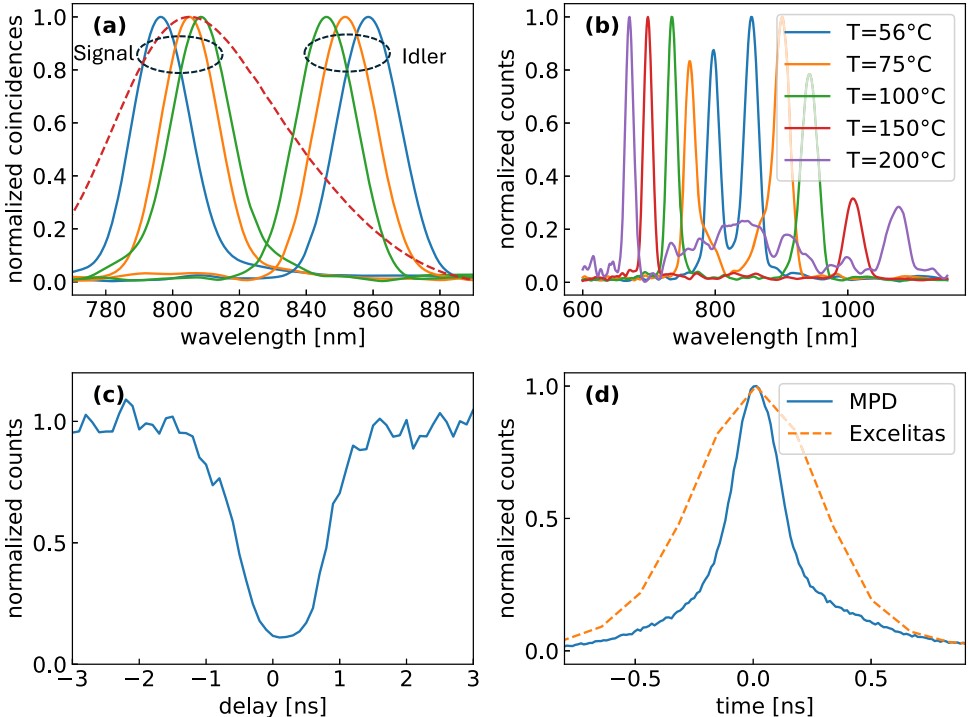

**Fig. 2 | Generation of frequency-tunable entangled photon pairs. a** dashed line - 'singles' measurement of overall signal spectrum generated by the ppKTP crystal at 56 °C, measured with a TWINS interferometer in front of the SPAD. The spectral correlations allow to select different signal spectra (solid lines, grouped under the 'signal' label) by placing a tunable interference filter on the herald arm (solid lines, collected under the 'herald' label); **b** SPDC spectra generated by the ppKTP crystal for different temperatures, showing tunability from 670 nm to 1100 nm; **c** $g^{(2)}$ measurement of the signal pulses performed with an HBT interferometer; **d** IRF of the system measured detecting coincidences between signal and idler pulses, with two SPADs from MPD and Excelitas. The FWHM of the response is 260 ps (MPD) and 600 ps (Excelitas) respectively. The correlation times measured by TCSPC are limited by the IRF of the detectors and are significantly longer than the intrinsic bi-photon correlation times that arise from SPDC.

## Results

We first characterize the source of EPPs by placing the second SPAD on the signal beam path. Figure 2a shows the 'singles' spectrum (dashed line) of the signal detected with the TWINS for a temperature T = 56 °C of the ppKTP, which is broadband because the SPDC process is close to the degenerate condition, with group velocity matching between signal and idler[33]. By placing a spectral filter that can be angle-tuned from 840 nm to 860 nm on the herald photon detector, we can selectively measure only the fluorescence photons that were excited by signal photons that have a conjugate wavelength to the herald photon, i.e. 810 nm to 800 nm in this case (solid lines). Under these conditions, the number of coincidences was ~2 × 10⁵/s.

The phase matching curves for signal and idler can be tuned out of the degeneracy condition by heating the ppKTP crystal, obtaining complete tunability of the signal from 670 nm to 1100 nm (measurements shown in Fig. 2b). Even broader tunability towards the green/blue could in principle be obtained by using a crystal with multiple poling periods. One should note that we demonstrate broad tunability in the excitation wavelength by using a very simple system, which consists of a CW laser and a periodically poled crystal. This should be compared to the considerably more complex systems, such as a mode-locked oscillator pumping an optical parametric oscillator[34], used for time-resolved fluorescence spectroscopy with classical light.

To check that the light exciting the sample consists of single photons, we characterized the signal beam with a heralded Hanbury-Brown-Twiss (HBT) interferometer, consisting of a beam splitter coupled to two SPADs in transmission and reflection, respectively, as shown in Fig. 1c. Figure 2c shows the normalized second order correlation function $g^{(2)}$ as a function of the delay between the detector readings, conditioned to the arrival of the idler photon; it displays a clear dip at zero delay indicating single-photon emission, which

becomes deeper approaching zero upon reducing the pump power to the ppKTP crystal.

Finally, we determine the time resolution and the instrumental response function (IRF) of the experiment by recording the coincidences as a function of time. Since the temporal correlation of the EPPs is much shorter than the response time of the SPADs, we can consider that the IRF is exclusively determined by the SPADs. For this experiment we used two types of SPAD: one with smaller area (50 μm²) but faster response time (MPD PD-050-CTD-FC) and one with larger area (180 μm²) and higher quantum efficiency but slower response time (Excelitas SPCM-800-14-FC). While the faster SPAD is always used to detect the heralding photon, which can be tightly focused due to its high beam quality, a trade-off between signal strength and response time occurs for the signal photon which, due to the incoherent nature of the fluorescence, is more challenging to refocus. Figure 2d compares two IRFs measured using the fast and the slow detector on the signal photon. Overall, upon deconvolution of the IRF, the temporal resolution of our measurements ranges between 100 and 200 ps. We then send the signal photon to the sample and detect the fluorescence using the MPD SPAD. Figure 3a shows a wavelength-integrated time-resolved fluorescence trace recorded for the infrared dye 800CW in dimethyl sulfoxide (DMSO) solvent. One observes a mono-exponential decay with a time constant $\tau = 1.51 \pm 0.01$ ns, in excellent agreement with classical TCSPC measurements obtained upon photoexcitation with a ~100-fs pulse at 800 nm. Similarly, Figure 3b shows the time-resolved fluorescence for the dye IR143, which displays a significantly faster decay with time constant $\tau = 0.79 \pm 0.01$ ns. The uncertainty reported in the lifetime values refer to the standard deviation of the fits and relates to the precision with which the lifetime is measured.

The acquisition time for the traces reported in Fig. 3a, b is 30 s. One should note that TCSPC using EPPs generated by a CW laser was

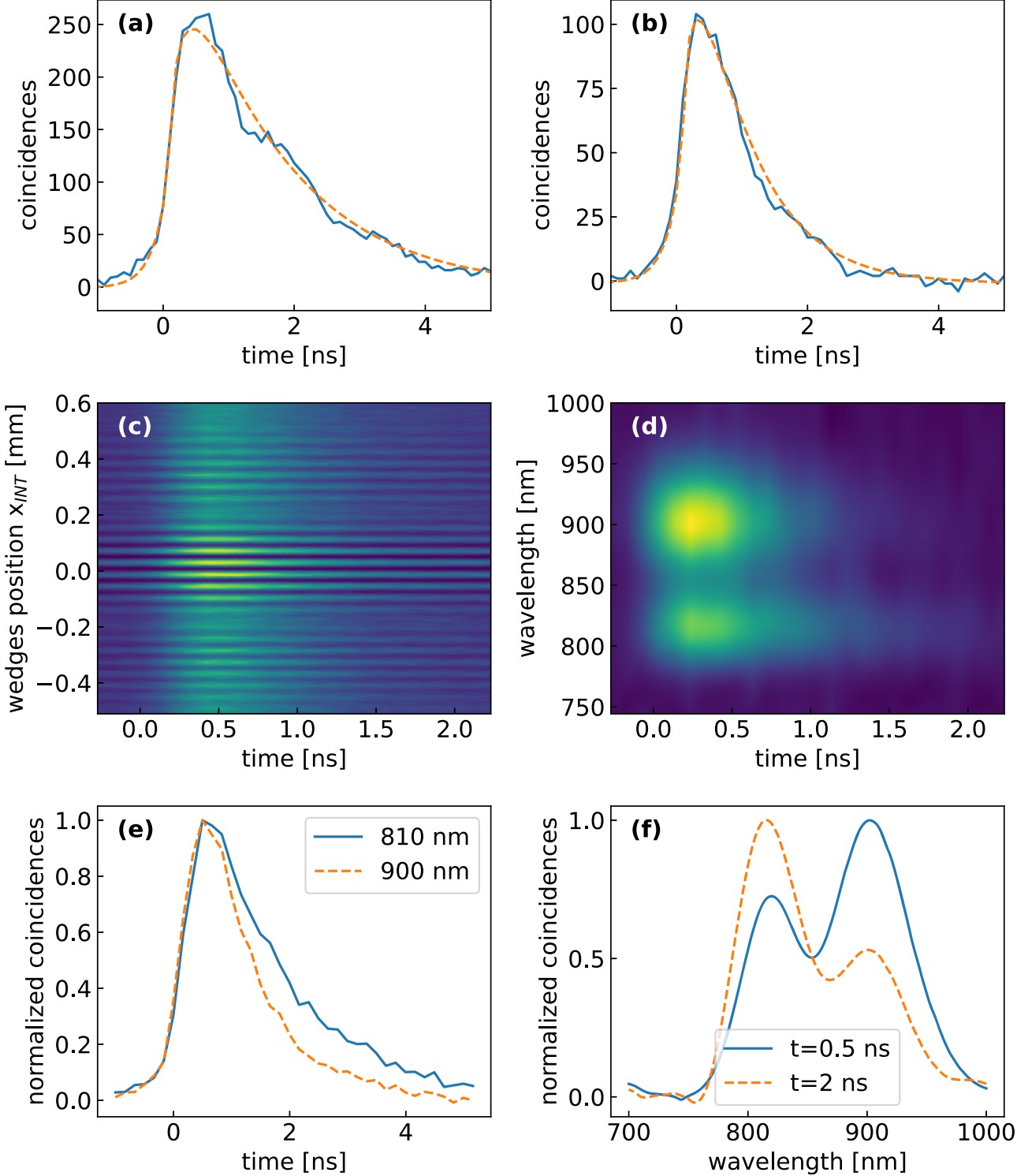

**Fig. 3 | Time- and frequency-resolved fluorescence spectroscopy at the single-photon level. a** wavelength-integrated fluorescence decay from 800CW infrared dye in DMSO following single-photon excitation at 800 nm, and mono-exponential fit with $\tau = 1.51 \pm 0.01$ ns (dashed line); **b** wavelength-integrated fluorescence decay from IR143 infrared dye following single-photon excitation at 800 nm, and mono-exponential fit with $\tau = 0.79 \pm 0.01$ ns (dashed line); **c** 2D fluorescence map as a function of emission time and wedge position $x_{INT}$ of the TWINS interferometer for a mixture of 800CW and IR143 dyes in DMSO solvent, following single-photon excitation at 800 nm. **d** time-dependent fluorescence spectra obtained by a Fourier transform of map (**c**) with respect to $x_{INT}$; **e** fluorescence decay dynamics at selected wavelengths; **f** fluorescence spectra at selected delays.

recently reported[35–37] however, with significantly lower temporal resolution and longer measurement time. Our results demonstrate that TCSPC using EPPs can be performed with similar data quality, in terms of temporal resolution and signal-to-noise ratio, to the classical measurement and without significantly compromising the

measurement time, while dramatically decreasing the excitation fluence.

To demonstrate the power of our approach to characterize spectro-temporal dynamics, we measure a mixture of the two dyes 800CW and IR143, dissolved in the solvent DMSO. Both dyes absorb at

800 nm but display different fluorescence spectra, peaking at 810 and 900 nm respectively, and lifetimes (see above). To obtain resolution in the detection frequency, we insert the TWINS interferometer on the detection path and record a fluorescence decay trace for each position $x_{INT}$ of the wedge generating the replicas in the interferometer, obtaining the 2D map shown in Fig. 3c. To compensate for the insertion losses of the interferometer, we used the Excelitas SPAD for this measurement, trading temporal resolution for photon collection efficiency. The total measurement time is 120 min due to the need to record multiple time traces, one for each wedge position $x_{INT}$. By performing a Fourier transform with respect to $x_{INT}$ and applying a suitable calibration procedure described in detail in Ref. 38, we obtain the time- and frequency-resolved fluorescence map reported in Fig. 3d. This shows that our approach allows one to obtain a remarkable amount of information on the spectro-temporal dynamics of the system following single-photon excitation. The time- and frequency-resolved fluorescence map of the mixture shown in Figure 3d displays two peaks, corresponding to the two dyes, with different lifetimes, see Fig. 3e. As illustrated in Fig. 3f, the shape of the fluorescence spectrum thus evolves in time, peaking at 910 nm (820 nm) at 0.5 ns (2 ns).

We then characterized light-harvesting complexes of the photosynthetic purple bacterium *Rhodoblastus (R.) acidophilus*. The photosynthetic membrane of a purple photosynthetic bacterium[39,40] contains peripheral antenna complexes known as LH2 complexes, which surround the core complex (LH1), containing at its center the reaction center (RC) in which charge separation takes place. As discussed in the introduction, the LH2 complex contains the B800 and B850 BChls, with absorption of the $Q_y$ transitions peaking at 800 and 850 nm, respectively. After excitation at 800 nm, EET to B850 takes place on the picosecond time scale, followed by a nanosecond fluorescence from B850 which is the terminal emitter[30]. The LH1 complex also contains another larger ring of BChls, called B875, with absorption red-shifted to 875 nm, thus ensuring downhill EET and energy capture from the peripheral complexes. From B875 a further EET to the RC takes place, where efficient charge separation occurs (see Fig. 4a). Figure 4b shows a time- and frequency-resolved fluorescence map of the LH2 from *R. acidophilus* in buffer solution, following single-photon excitation at 800 nm. As expected, and in agreement with[29], we observe a mono-exponential decay of the fluorescence from B850. Our time resolution is in fact too low to observe the ~1-ps EET from B800 to B850. We do not observe a significant wavelength dependence of the decay time. Figure 4c shows a cut at 860 nm and reveals a time constant of $1.13 \pm 0.01$ ns, in excellent agreement with classical measurements performed with a femtosecond pulsed laser (also shown in the figure). We then moved to entire photosynthetic membranes containing both LH2 and LH1. In particular, the RCs in LH1 can be either open (i.e. functional and ready to perform charge separation) or closed (i.e. with the charge separation process blocked). Figure 4d shows a wavelength integrated fluorescence dynamics for a photosynthetic membrane with open RCs. In this case, the EET processes from B850 to B875 and from B875 to the RC shorten the excited state lifetime to $101 \pm 3$ ps. When instead the RCs are closed (by inserting the reducing agent sodium dithionite in the buffer) the charge separation process is inhibited and the RC equilibrates with the LH1, which becomes the terminal emitter. In this case, the fluorescence lifetime increases to $248 \pm 9$ ps. Figure 4e shows the time-integrated spectra for open and closed RCs, measured with the TWINS interferometer. When the RCs are closed, the fluorescence red-shifts, peaking at 875 nm due to the superposition of emission from B850 and B875 BChls. When the RCs are open, the charge separation process in the RCs quenches the fluorescence from B875 nm so that the residual emission, dominated by B850, is blue-shifted.

Finally, to show the broad applicability of our approach to time-resolved spectroscopy using EPPs, we demonstrate its capability to record photoinduced dynamics without significantly compromising

the measurement time with respect to classical approaches. Figure 5 shows wavelength-integrated TCSPC time traces of LH2 for various measurement times: (a) for 50 s the SNR is very high, ~34, then it progressively degrades with shorter acquisition times ((b) and (c), 10 and 2 s, respectively) with a still acceptable SNR ~4 at 0.6 s acquisition time (d) that provides a lifetime estimate in-line with estimates obtained at the longer acquisition times.

## Discussion

In this work we show how quantum correlations can be used to perform time-resolved spectroscopy of ultrafast photoinduced processes under the least possible perturbative conditions. We overcome the paradigm of classical time-resolved spectroscopy, which is performed using pulsed light sources, and start from a continuous wave laser exploiting the ultrashort correlation time between randomly generated signal/idler pairs to obtain temporal resolution. Importantly, we demonstrate that time-resolved spectroscopy with quantum light can be performed without compromising measurement time, recording a fluorescence time trace in less than a minute, that can be reduced to ~1 s with acceptable signal-to-noise ratio. We also show the ability to add spectral resolution to the measurement, using a Fourier transform approach which employs an interferometer on detection. While our temporal resolution is currently limited to ~100–200 ps by the instrumental response function of the photodetectors, it could be improved by up to one order of magnitude using suitable detectors, such as superconducting nanowire single-photon detectors. The time resolution could be further improved to ps or sub-ps levels by exploiting the Hong-Ou-Mandel (HOM) effect, whereby the fluorescence photon is made to interfere with an identical photon, also generated by SPDC[41]. Since the photon correlation time (related to the phase matching bandwidth of the SPDC process) can be as short as hundreds of femtoseconds, one should be able to achieve sub-picosecond temporal resolution by measuring the rise and decay time of the HOM dip.

We show improvements in acquisition times and temporal resolution that represent a step change in terms of capability compared to previous work and transform the use of SPDC for lifetime sensing into a relevant experimental tool. It is also this improvement that allows us for the first time to simultaneously measure the lifetime and emission spectrum (i.e. spectrally resolved lifetime). Beyond this, we also take things one step further in terms of exploiting the correlations of SPDC —the herald photon is filtered so as to ensure that we collect fluorescence only from events triggered by pump (signal) photons with the desired frequency. We are therefore conditioning our fluorescence photon measurements simultaneously in time and frequency (of the pump photon).

In summary, we are not exploiting entanglement directly in the sense of violating classical sensitivity or resolution bounds that might arise from entanglement nor does our approach demonstrate a direct impact of quantum properties on the resolution or SNR. However, we do show that correlations from entangled photons can be used for efficient and fast detection of fluorescence lifetime signals with a CW laser source. This cannot be achieved by classical means. The sub-second acquisition times using SPDC photon pairs also lay the ground for future work and development of systems that do exploit quantum properties, such as HOM interference for picosecond or sub-picosecond lifetimes but now starting from experiments that can be carried out at rates or acquisition times that are of interest to biologists.

## Methods

### Fluorescent dye sample preparation
The dyes IR143 (perchlorate crystalline powder, from Exciton) and 800 CW (NHS Ester, Lycorbio) were dissolved in DMSO (purity 99%, Sigma-Aldrich) at a concentration of ~1.5 mmol L$^{-1}$, similar to the work e.g.

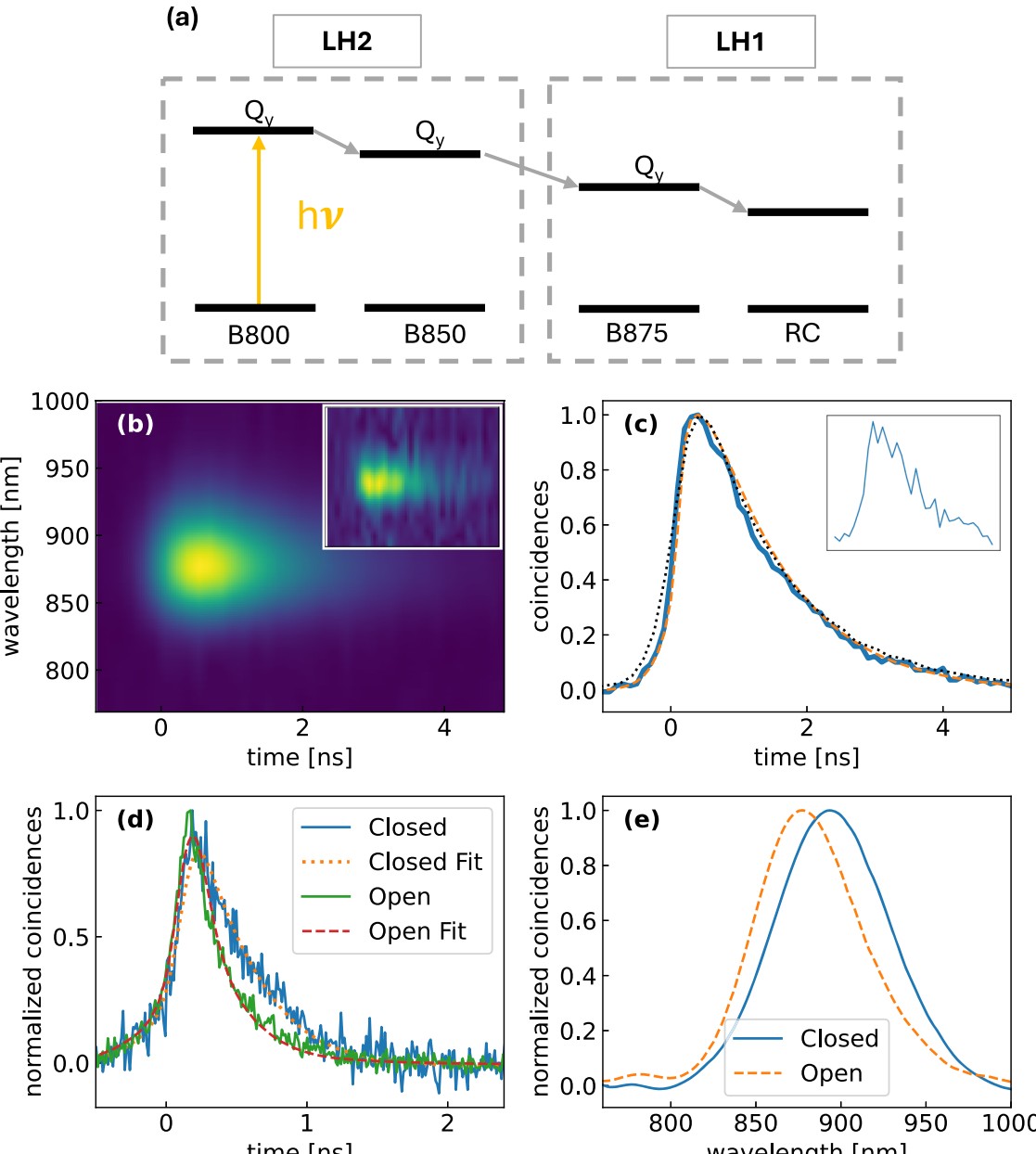

**Fig. 4 | Time- and frequency-resolved fluorescence spectroscopy of photosynthetic membranes at the single-photon level. a** Energy level scheme of the BChls in the LH2 and LH1 complexes, indicating the cascade of EET processes leading to light capture in the RC and charge separation; **b** time and frequency-resolved fluorescence spectrum of LH2 of the purple photosynthetic bacterium *R. acidophilus*, following single-photon excitation of B800 BChls and EET to B850 taken with 200 interferometer scan points, 1 min each (total acquisition time, 3.3 h). The inset shows the same measurement, retaken with 100 scan points, 2 s dwell time, 3.3 min total acquisition time; **c** fluorescence decay at 860 nm taken from (**b**) after integrating along the wavelength axis) and fit (dashed line), blue continuous line single photon excitation, black dotted line femtosecond pulse excitation (the inset shows the decay taken from the inset to (**b**)); **d** wavelength-integrated time-resolved fluorescence decay of the photosynthetic membrane of *Rhodobacter sphaeroides*, following single-photon excitation at 800 nm with open (blue trace) and closed (red trace) RCs; **e** time-integrated fluorescence spectra of the photosynthetic membrane with open (blue) and closed (orange dashed) RCs.

in[37]. For the measurements, 250 μL of each dye solution was transferred into separate 1 mm path-length quartz cuvettes (CV1Q035AE, Thorlabs).

To prepare the mixed sample, 75 μL of the 800CW solution and 150 μL of the IR143 solution were combined in a third cuvette.

## Light harvesting complexes preparation

The LH2 complexes from the purple photosynthetic bacterium *Rhodoblastus acidophilus* strain 10050 (formerly known as *Rhodopseudomonas acidophila*) were prepared as previously described by Cogdell et al.[42]. The purified LH2 complexes were suspended in 0.1% v/v LDAO

in 20 mM Tris HCL pH 8.0. Photosynthetic membranes from *Rhodobacter sphaeroides* strain 2.4.1 were prepared from cells grown anaerobically in the light as previously described by Cogdell et al.[43]. The purified membranes were suspended in 20 mM MES pH 6.3, 100 mM KCl. The reaction centers in the membranes were 'closed' by the addition of sodium dithionite, which reduced the reaction center's primary electron acceptor (as described in[43]). Photosynthetic membranes from *Rb. sphaeroides* were used, rather than those from *R. acidophilus*, because they are small membrane vesicles (chromatophores) that are much less light scattering than the highly light scattering membrane sheets that are produced by *R. acidophilus*. Both

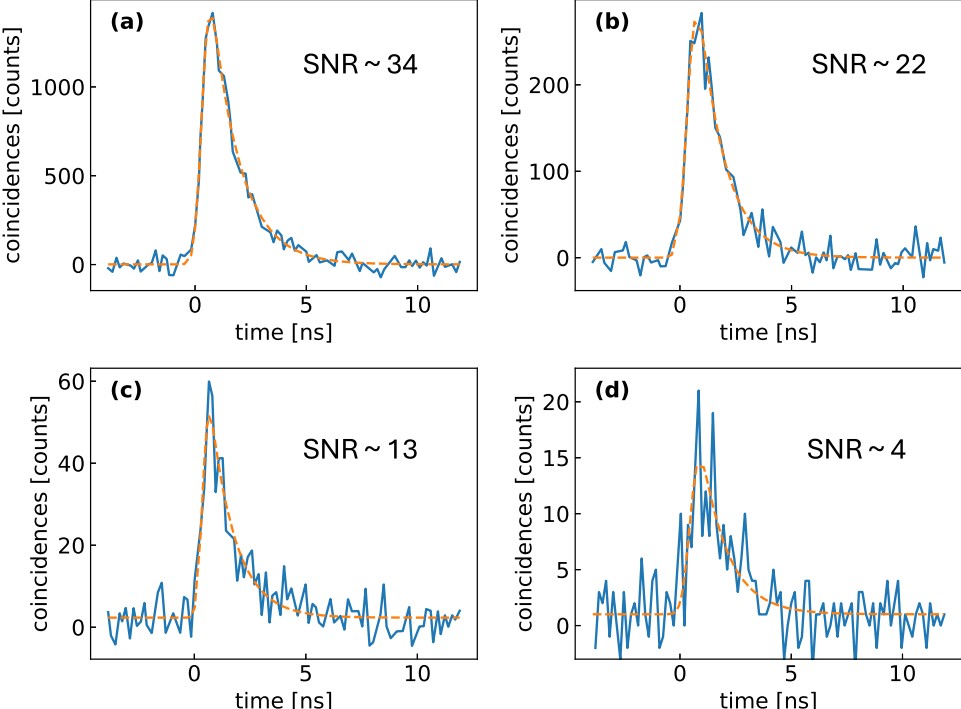

**Fig. 5 | LH2 lifetime estimation for various integration times.** Wavelength-integrated fluorescence decays (total counts) for LH2 of the purple photosynthetic bacterium *R. acidophilus*, following single-photon excitation at 800 nm, measured with different integration times: (**a**) 50 s ($\tau = 1.14$ ns); (**b**) 10 s ($\tau = 1.16$ ns); (**c**) 2 s ($\tau = 1.11$ ns); (**d**) 0.6 s ($\tau = 1.17$ ns). Orange dashed lines are exponential fits convoluted with the IRF. SNR values are indicated in each graph and are estimated as SNR = ⟨counts⟩/⟨σ⟩, where ⟨·⟩ indicates the average estimated from the the 10 points around the highest intensity point and σ is the standard deviation of the residuals.

species have very similar complements of LH complexes. Before measurements, 200 μL of these suspensions were transferred to individual quartz cuvettes of 1 mm path length (CV1Q035AE, Thorlabs).

## TWINS interferometer

TWINS (manufactured by NIREOS s.r.l.) consists of two plates of a birefringent material, with thicknesses $L_a$ and $L_b$ respectively, and optical axes rotated by 90°. For an incident optical waveform with polarization rotated by 45° with respect to the ordinary and extra-ordinary axes, the output waveform splits into two replicas with perpendicular polarizations and delay $\tau$ proportional to $L_b - L_a$. To continuously change the delay, one of the plates is cut into a pair of wedges, one of which is transversely translated. The two delayed replicas are then made to interfere by projecting them onto the same polarization with a polarizer. As it is a common path interferometer, TWINS guarantees exceptional delay stability and reproducibility.

## TWINS interferometer and signal-to-noise ratio

We compare the SNR for the case of the TWINS coupled with a single-pixel SPAD and a grating spectrometer coupled to a camera.

The SNR for the (TWINS) single pixel case is

$$\text{SNR}_{\text{pixel}} = \frac{S \cdot T}{\sqrt{S + D_{\text{pixel}}T}} = \frac{\sqrt{T} \cdot S}{\sqrt{S + D_{\text{pixel}}}} \quad (2)$$

where S = signal, D = dark counts, T = measurement time. Compared to the TWINS-single pixel case, the grating case leads to reduction of S to S/m. The reduction factor m = N/q contains two terms: N is the number of pixels on the camera, q is the (typical) reduction in photon detection probability in the SPAD array compared to a single pixel SPAD. The reduction q arises as a result of the lower fill factor of the active pixel area in SPAD arrays (of the order of 1%–50%) compared to single pixels

(100%) and reduced quantum efficiency in SPAD arrays (max typically around 50% but as low as 1–10% at 800 nm) compared to a single pixel (as high as 70%–80%, also at 800 nm). We could also account for grating efficiency but if we take this to be 50% then we can ignore this loss as we have a similar loss also for the TWINS as this requires polarized light at the input.

For a grating coupled to a SPAD camera we have

$$\text{SNR}_{\text{camera}} = \frac{\sqrt{T}(S/m)}{\sqrt{S/m + D_{\text{camera}}}} = \frac{\sqrt{T} \cdot S}{\sqrt{S + D_{\text{camera}}m}} \quad (3)$$

The ratio of the SNRs for the same signal integration time T is then:

$$R = \frac{\text{SNR}_{\text{pixel}}}{\text{SNR}_{\text{camera}}} = \frac{\sqrt{S + D_{\text{camera}}m}}{\sqrt{M}\sqrt{S + D_{\text{pixel}}}} \quad (4)$$

where we have accounted for the fact that we must also scan the TWINS spectrometer by $M$ steps (typically 100 or 200 scan points are used in our experiments), thus leading to a factor $\sqrt{M}$ in favor of the spectromer+SPAD camera approach (where we can acquire for a factor M longer in time).

We note that R does not depend on time as both the signal and the dark counts grow linearly with time and this dependence cancels out. We can provide estimates by inserting values based on our own measurements in 1 s with S ~ 100 counts. For our single SPADs we have $D_{\text{pixel}}$ ~ 50 counts/second and 70% efficiency (Excelitas SPSM-NIR) and if we take a linear SPAD array so as to maximize fill factor (Pi-Imaging, SPADlambda) we have 320 pixels with $D_{\text{camera}}$ ~ 250 counts/second and q = 0.08 at 800 nm (80% fill factor, 10% efficiency), i.e. m = 4000, we then have R = 8. As a further example, one might instead consider another state-of-the-art camera, e.g. SPAD512 (PI Imaging), 512 pixels, $D_{\text{camera}}$ ~ 25 counts/second, q = 0.05 at 800 nm (50% fill factor, 10%

efficiency), i.e. m = 10240 at 800 nm: R = 4. These estimates based on actual values of available technology indicate that for the same total acquisition time, the TWINS (coupled to a single SPAD pixel) SNR outperforms a grating spectrometer with a SPAD array.

## Data availability

All data used to produce the plots shown in this article is available at https://doi.org/10.5525/gla.researchdata.1956.

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

## Acknowledgments

The authors acknowledge financial support from the Royal Academy of Engineering under the Chairs in Emerging Technology and Research Fellowships schemes and the U.K. Engineering and Physical Sciences Research Council (Grants No. EP/X035905/1, EP/Y029097/1, EP/Z533166/1). G.C. and L.U. acknowledge support by the Progetti di ricerca di Rilevante Interesse Nazionale (PRIN) of the Italian Ministry of Research 2022HL9PRP Overcoming the Classical limits of ultRafast spEctroSCopy with ENtangleD phOtons (CRESCENDO). G.C. acknowledges funding from the European Union-NextGenerationEU under the National Quantum Science and Technology Institute (NQSTI) grant no. PE00000023-q-ANTHEM-CUP H43C22000870001.

## Author contributions

R.A. - methodology, investigation, formal analysis, visualization, writing, review and editing; L.U.- methodology, investigation, formal analysis, visualization, writing the original draft; A.L. - supervision, writing, review and editing; R.J.C. - resources, writing, formal analysis, review and editing; G.C. - conceptualization, supervision, funding acquisition, writing the original draft; D.F. - conceptualization, supervision, funding acquisition, writing, review and editing.

## Competing interests

G.C. discloses financial association with the company NIREOS, which manufactures the TWINS interferometer used in this paper. The remaining authors declare no competing interests.
