## [Transparent Peer Review file · Nature Communications]

Correlated-photon time- and frequency-resolved optical spectroscopy

Corresponding Author: Professor Daniele Faccio

Version 0:

Reviewer comments:

Reviewer #1

(Remarks to the Author)

Please find attached the review report.

Reviewer #2

(Remarks to the Author)

This manuscript presents the first experimental realization of time-resolved fluorescence spectroscopy with entangled photon pairs. Although several theoretical works have emphasized the advantages of entangled light for time-resolved spectroscopy, an experiment had not previously been carried out—primarily because the extremely weak nonlinear signals were expected to make such measurements impractical. By combining fluorescence detection with a TWINS interferometer, the authors now demonstrate that these measurements are indeed feasible. While the spectral information obtained in this measurement does not exceed what can be accessed with state-of-the-art time-resolved laser spectroscopy, the study is nonetheless highly significant: it lifts time-resolved spectroscopy with entangled photons from theoretical proposal to proven experimental technique. Provided the comments below are addressed, I believe the manuscript is well suited for publication in Nature Communications.

[Comment 1]

Please add a brief explanation of why the PBS placed next to the objective in Fig. 1 (a) is used instead of a dichroic mirror. I assume the PBS ensures that horizontally polarized light enters the subsequent TWINS interferometer for further polarization control, but researchers performing experiments with microscopes would normally employ a dichroic mirror at this position. A short clarification would therefore make the setup easier to understand.

[Comment 2]

Please plot the raw photon counts in Fig. 5 instead of normalizing the data to unity. Absolute count rates allow experimentalists to judge how demanding the measurement is and to assess the practicality of reproducing the experiment.

[Comment 3]

Several key theoretical papers on time-resolved spectroscopy with entangled photons are missing from the Introduction:

[A] A. Ishizaki, J. Chem. Phys. 153, 051102 (2020).

[B] L. Ko, R. L. Cook & K. B. Whaley, J. Phys. Chem. Lett. 14, 8050 (2023)

[C] J. J. Fan, Z. Y. Ou & Z. Zhang, Light Sci. Appl. 13, 163 (2024)

[D] Y. Fujihashi, O. Iso, R. Shimizu & A. Ishizaki, arXiv:2502.02073 (2025)

In particular, while reference [D] treats SPDC driven by a pulsed laser rather than by a continuous-wave source, both studies are conceptually similar in that they exploit entangled photon pairs to obtain time-resolved fluorescence spectra. The authors should therefore comment this connection in the manuscript.

Reviewer #3

(Remarks to the Author)

Alvarez-Mendoza and co-workers describe an experimental technique exploiting pairs of entangled photons generated by a CW laser source to realize time and frequency resolved TCSPC measurements.

The technique is applied to two example applications, claiming an advantage in the signal-to-noise versus typical integration times for similar measurements demonstrated in the past.

The combination of reasonably short integration times with the use of a CW illumination scheme is appealing in terms of overall cost and simplicity of the experimental apparatus. Frequency resolved measurements are also presented, even though performed at the cost of a linear increase of the measurement time.

The manuscript is a timely contribution dealing with an interesting topic. The idea for this work is somehow indebted to a previous paper which confirmed that energy transfer in photosynthetic antenna complexes works on a single-photon basis. This work is correctly acknowledged in the manuscript, with a clear indication of how the current work advances over the previous one. On the other hand, the novelty and technological advantage of the current implementation over previously demonstrated ones [31-33] is not clearly discussed nor quantified. With respect to previous works, this manuscript adds frequency resolution, however at the cost of repeating several measurement to reconstruct spectral information. This may be (trivially?) possible also with previously shown implementations, suggesting the lack of an inherent advantage in the proposed measurement scheme. For publication on Nature Communications, it is important to show that the newly proposed method is not incremental with respect to previous results.

In general, the Authors should clarify:

- how much of the signal-to-noise ratio enhancement is provided by the CW illumination scheme (which is quite recent, but not new), how much by the quality of the detection scheme (which relies on commercial detectors), the NA of the objective, or the exploitation of the quantum nature of the photon pair (besides their temporal/frequency coincidence). In terms of detection, I understand that part of the signal-to-noise improvement ratio comes from the decision to use SPAD detectors rather than SPAD arrays. On the other hand, using a single SPAD detector pushes the duration of spectral measurements into the hours. How does this compare to the previous literature?
- If the observed advantage comes mainly from the detection scheme, I wonder whether it could be translated to other "classical" techniques. Pump-probe and optical-gating measurements can also work with very low probe intensities (despite requiring still pulses operation).
- Spectral resolution is provided by the TWINS device, which however introduces also insertion losses and consequently a trade-off on the temporal resolution. Can the Authors comment on the possibility, advantages and disadvantages of using instead dispersive elements, spectral filters, etc. as alternative ways of achieving spectral information? Would there be any specific penalty, considering that the spectral information is still reconstructed based on several (how many?) sequential measurements?
- In their introduction, the Authors present some possible strategies for a quantum advantage, such as the possibility of overcoming "granular (shot) noise" or to "reach the so-called Heisenberg limit". However, it is not clear whether either of these two quantum-specific enhancements are demonstrated in this work, or whether only the temporal/frequency correlation is used.
- The comparison against the results already presented in references [31-33] should be made more quantitative. The two aspects of signal-to-noise ratio vs integration time, and that of temporal resolution should be discussed separately. Regarding signal-to-noise ratio and the uncertainty in the determination of decay lifetimes reaches a level of 0.01 ns within about ~100 s in these previous papers. The Authors claim similar uncertainty levels with shorter integration times between 30-50 s, leading to a possible improvement of a factor 3-4 with respect to these previous reports. However, it is unclear what the uncertainty of 0.01 ns actually represents in the measurements shown by the Authors, since other descriptors of the fitting goodness are missing. Such a low uncertainty seems also not compatible with the IRF deconvolution step, coarseness and noise level of the curves shown. This is a critical factor as the careful determination of this uncertainty underpins the main claim of this work. Previous works provided a more quantitative overview on these aspects, rather than just showing qualitatively the noise level at different integration times.
- Similarly, in terms of resolution limits, some of the previous experimental works cited by the Authors claimed to be theoretically able to measure decay times shorter than those measured experimentally in this work. Hence, the key claim that previous reports showed "significantly lower temporal resolution and longer measurement time" does not appear to be sufficiently substantiated.
- In the comparison against the results in [31-33], another important point which is not commented upon in this work is that of the dye concentration. Signal-to-noise and integration time depend critically on the molar concentration, and integration times should be further normalized by this quantity. These aspect should also be taken into account when comparing the results to those in [31-33].
- Finally, I have a couple of comments on the current wording of the abstract, which is potentially misleading. A statement such as "we overcome this paradigm by exploiting quantum correlations to perform time-resolved spectroscopy with entangled photons" seems to claim a level of novelty that was already demonstrated in previous works (perhaps the sentence can be removed altogether without altering too much the sense of the abstract). Similarly, the abstract mentions twice the fact that "experiments are performed under conditions comparable to real-world sunlight illumination". Since solar light is not entangled, this statement refers solely to average intensity levels (impinging on the non-linear crystal? or interacting with the sample?), or would this be a hint at the change of generating entangled photon pairs using solar light on the crystal
- "t = 56" should be "T = 56°C"

In conclusion, the Authors need to make their work more quantitative to support their claims and explain whether these represent a more a scientific or rather a technological advancement, especially when compared to the results already present in the literature.

Pending suitable revisions to clarify these points, the paper can be reconsidered for publication at Nature Communications.

Reviewer #4

(Remarks to the Author)

Review for "Entangled-photon time- and frequency-resolved optical spectroscopy"

This manuscript follows a recent paper in Nature [Ref 25, Li et al.] showing by fluorescence measurements that the excitation process underlying photosynthesis occurs under illumination with heralded single photons. This provides evidence that photosynthesis is a single-photon process. This manuscript performs a similar experiment, illuminating a sample of photosynthetic bacteria with the signal beam from a non-degenerate parametric down conversion source. By performing time-resolved single-photon measurements on the idler field, the fluorescence decay curve in the signal arm can be measured with reasonable precision. The authors claim several times that this is a case of "exploiting entanglement" between the signal and idler photons.

In my opinion, however, entanglement plays no role in this experiment. Correlations between the timings of the signal and idler photons certainly do play a role, but this is not adequate to say that the experiment "exploits entanglement". Entanglement is a coherent effect that requires measurement basis changes to be made in both fields to manifest, but here the idler photon is only ever subjected to direct time-resolved detection. The function of the idler field, as far as this experiment goes, is therefore simply to convey classical timing information about when the single-photon signal pulses are emitted.

Most of the results of this experiment, in my view, could be replicated with ultrafast classical pulses with the same bandwidth as the emission spectrum of the SPDC source. The exception is the $g(2)$ measurement of the heralded signal photon, but this does not impact on the other results of the paper, which are not in any way dependent on the single-photon character of the signal field.

The authors also perform a spectrally resolving measurement on the signal field using a Fourier-transform spectroscopy, implemented by a common-path polarization interferometer inserted into the path of the fluorescence. By splitting the pulse into two copies separated by a variable delay, some timing information is lost (in the sense that more measurements are required to estimate the arrival time), with the payoff that the transmission probability of the photon carries some information about the spectrum. However, many copies are required to obtain a full interference pattern needed to recover the spectrum. FTS is already very widely used so I don't think this is an especially novel inclusion. It is also not mentioned that the TWINS interferometer also introduces on average 50% loss at the polarizer.

A few other related criticisms:

The authors say: "We can nonlocally control the spectrum of the excitation photons that lead to detected fluorescent photons by placing on the idler an interference filter with 10-nm bandwidth at 860 nm." It is inaccurate to describe this as any kind of "nonlocal control" as the marginal properties of the signal field are unchanged by locally filtering the idler. Rather, by choosing which photons can trigger heralding events, simply a different subset of signal photons give rise to coincidences. I feel that some of the introduction is not relevant. The authors emphasize in the first line of the abstract that most experiments in ultrafast spectroscopy are carried out with much higher intensities than are found in nature, before claiming to "overcome this paradigm by exploiting quantum correlations". They elaborate in the introduction that "[r]educing the excitation intensity to more realistic values would result in vanishing nonlinear signals for classical time-resolved experiments," which is true for nonlinear/multi-photon interactions. This is not relevant to the present paper, though, which is investigating a single-photon excitation process that is linear in the intensity. In fact, the key result of Li et al is that for precisely this process, the fluorescence signal from truly single-photon excitation is not only non-vanishing, but measurable; and indeed many other experiments have carried out spectroscopy at the single-photon level.

In summary, I therefore feel that there are overarching scientific weaknesses in this paper as written and it is not suitable for publication.

Version 1:

Reviewer comments:

Reviewer #1

(Remarks to the Author)

The authors have addressed all my comments, and I am satisfied with most of the answers. I think there are still two questions that have to be clarified.

1. In Fig.1a as added, it shows that the spectrometer/grating is involved in i-photon detector, to implement the frequency resolution. This seems confusing to me--I would assume the spectrometer is equipped within the s-photon detector, whereas the i-photon detection offers a temporal gate within a narrow time window. So which detector includes the spectrometer? The authors should clarify.

2. The authors should better make an estimation of time-frequency scale achieved in present work, i.e., in terms of $\Delta \omega * \Delta t$.

Reviewer #2

(Remarks to the Author)

The authors have thoughtfully addressed my questions. I believe the paper is suitable for publication.

Reviewer #3

(Remarks to the Author)

The Authors have comprehensively addressed all the points raised, which resulted in a significant improvement of the manuscript clarity and relevance. I therefore recommend the manuscript for publication without further amendments.

Reviewer #4

(Remarks to the Author)

The authors address some of my points, and I am happy to see they have revised some of their claims in response to my comments. My fundamental criticism remains however: the paper pitches itself as a demonstration of a new type of measurement where quantum entanglement plays a fundamental role, which I do not agree is the case.

I do not agree with the authors' claim that just using the spectral or temporal correlations individually from an SPDC source are 'routinely summarised in literature by saying that we "exploit entanglement"' - at least, not correctly. The considerable interest in entanglement-enhanced spectroscopy is driven by the promise of a coherent boost in sensitivity that is predicted from entangled photon pairs, such as in multi-photon spectroscopy. Nothing like this is demonstrated here. The timing precision here is obtained by the temporal correlations in the photon pairs emitted by parametric fluorescence, but there are many classical sources that also exhibit temporal correlations, and pulsed sources can be trivially correlated with an electronic gating signal. The correlations present in SPDC are in a sense an artifact of the quantum nature of the source, but the underlying entanglement itself is not exploited here in any meaningful sense. In response to mine and Reviewer 3's comments the authors have somewhat clarified this with the text added towards the end of p1. But even after these revisions, there is still a strong risk that readers might misinterpret this work as claiming to demonstrate the coherence-boosted proposals that are the real subject of the current interest in quantum-enhanced spectroscopy. Unfortunately, this misunderstanding is already evident in some of the other referee comments.

In their responses the authors have also emphasised the spectral correlations in the fluorescence, but these are not especially important to the measurement. Spectral filtering of the idler and postselecting on coincidences is used to reduce noise from other wavelengths, but this is essentially equivalent to filtering the signal itself. Excessively narrow spectral filtering, in fact, will destroy the temporal correlations that are the basis of the paper due to the Fourier conjugacy of the spectral and temporal bandwidths. A true entanglement-exploiting measurement would necessarily involve a joint measurement on the signal and idler biphoton, such as Hong-Ou-Mandel interferometry.

The authors have pivoted to claiming that the advantage of the paper is in the technical expediency of using a CW scheme, harnessing the fact that even with a CW pump, parametric fluorescence contains both tight temporal and broadband spectral correlations. I note that in the original manuscript this was just a passing remark, but has assumed a much greater significance after their revisions. They accept that the use of a pulsed probe and temporal gating can reproduce the timing resolution, and hence the impact of the paper rests on the relatively greater availability and affordability of CW lasers and SPDC sources compared to pulsed lasers.

To end on a positive, I think this is a reasonable point. Indeed, there have been numerous other works in e.g. ghost imaging that do not claim quantum advantage per se, but take advantage of these natural correlations in parametric fluorescence as an experimental tool. A CW laser and nonlinear crystal is certainly cheaper and more convenient than a tunable picosecond laser, which makes this still useful without the questionable claims to be exploiting entanglement. For this reason I think there is some merit in this paper, but it still needs major revisions to dial down the claims to quantum advantage before it could be considered for publication. Even then, the idea is essentially methodological and I would be reluctant to recommend it for publication in such a high impact journal.

Reviewer #1

... I think the experiments performed by authors are a milestone in the journey towards the quantum-light ultrafast spectroscopy. Therefore the manuscript is of a great deal of publishing on Nat. Commun., after a major revision. Some of my comments are listed below.

We thank the reviewer for the appreciation of our work, for recognizing our results as “a milestone in the journey towards the quantum-light ultrafast spectroscopy” and for recommending its publication in Nature Communications. Below we reply in detail to the reviewer’s comments.

1. A schematic illustration is needed in Fig.1, in order to show the essence of how the entangled photons pump the molecules and temporally gate the dynamics, with the coincidence counting.

REPLY: We have added this as Figure 1(a) together with new text in the manuscript that explains the figure (first sentence in “experimental setup” section. The figure is reported below as Figure R1.

Figure R1: Time- and frequency-resolved spectroscopy with EPPs. (a) Schematic overview of spectral and temporal heralding of fluorescent photons. (b) Experimental setup. LP: long-pass filter; DM: dichroic mirror; PBS: polarizing beam splitter used to select a single input polarisation for the TWINS spectrometer; BP: bandpass filter; OBJ: microscope objective; SMF: single-mode fiber; MMF: multi-mode fiber; SPAD: single-photon avalanche diode; TCSPC: time-correlated single photon counting. (c) Setup for $g^{(2)}$ measurement. (d) Scheme of the TWINS interferometer. HW: halfwave plate; Plate: α -BBO plate; Wedges: movable α -BBO wedges; Pol: polarizer.

2. I am wondering if the intra-aggregate dynamics within LH2, i.e., between B800 and B850 could be resolved by the signal in Fig.4d, as the horizontal axis is of a scale ~ 1 ns. Since the temporal resolution by the entangled photons is < 200 ps, it would be better to provide the results for a shorter time scale, for example, < 1 ns. This will be very supportive for the conclusion in the manuscript.

REPLY: We thank the reviewer for this comment. The figure below shows the time-resolved fluorescence dynamics of LH2 zoomed on the 500-ps timescale. Unfortunately, the rise time of the signal is within the instrumental response function of the setup, which is ≈ 200 ps. It is therefore not possible to time resolve the B800 \rightarrow B850 excitation energy transfer process, which takes place with a much shorter time constant $\tau = 900$ fs (J. Phys. Chem. B **101**, 1087–1095 (1997)).

Figure R2: time-resolved fluorescence dynamics of LH2 zoomed on the 500-ps timescale.

3. The scheme in present work is like a “ghost imaging” in time and frequency domains (Well, the ghost imaging renders an elastic scattering of photons, but here the inelastic scattering dominates). The underlying mechanism of the time-frequency resolution in present work is that the fluorescence following the s photon pump has to pass a grating in detector 1 whereas the single-photon pulse in i arm provides the temporal gate; the single-photon detector for collecting i-arm photons does not need to contain the spectrometer. A spectrometer has to be integrated into the single-photon detector collecting the photons from s arm. I am wondering if my understanding is correct? Therefore the present work can go beyond the Nature one in 2023.

REPLY: we thank the reviewer for the accurate synthesis of our work. We believe that indeed temporal and spectral correlations between signal and idler photons are key to our results. We go beyond the Nature paper of 2023 in two important ways:

- i) We use a CW excitation laser to generate the EPPs, and exploit the temporal and also spectral correlation between signal and idler photons to achieve time resolution in the measurement of fluorescence;
- ii) We resolve not only the temporal dynamics but also the spectrum of the fluorescence photons. We thus perform a true time-resolved spectroscopy experiment upon single-photon excitation, as reflected in the title of our manuscript.

In the revised version of the manuscript, we have better clarified these concepts in the following sentence in the introduction:

“Whilst traditional time-resolved measurements are performed using ultrafast pulsed laser systems, here we demonstrate that by exploiting the frequency and time correlation properties of entangled photons, it is possible to perform high SNR measurements in short acquisition times using a CW laser.”

4. Have the authors characterized the spectral correlation of the entangled photons, without the samples? The authors should better provide this if yes. The anti-correlation in spectrum (or the correlation in time domain) does play a key role in achieving the time-frequency resolution beyond the Fourier limit.

REPLY: Yes, we have indeed characterized the spectral correlations. We now include a new figure 2a that shows both the heralding idler spectrum and the heralded signal spectrum for three different settings of the interference filter in front of the herald detector. We have also updated the text in the caption and manuscript to describe this. The new version of Figure 2 is reported below as Figure R3:

Figure R3: Generation of frequency-tunable entangled photon pairs. (a) dashed line - ‘singles’ measurement of overall signal spectrum generated by the ppKTP crystal at 56°C, measured with a TWINS interferometer in front of the SPAD. The spectral correlations arising from the SPDC entanglement allow to select different signal spectra (solid lines, grouped under the ‘signal’ label) by placing a tunable interference filter on the herald arm (solid lines, collected under the ‘herald’ label); (b) SPDC spectra generated by the ppKTP crystal for different temperatures, showing tunability from 670 nm to 1100 nm; (c) $g(2)$ measurement of the signal pulses performed with an HBT interferometer; (d) IRF of the system measured detecting coincidences

between signal and idler pulses, with two SPADs from MPD and Excelitas. The FWHM of the response is 260 ps (MPD) and 600 ps (Excelitas) respectively.

5. It is worth noting some works related to the present work [Light: Sci & Appls 13, 163 (2024); JCP 160, 104201 (2024); Light: Sci & Appls 11, 274 (2022)]. Pathway selectivity and timeenergy resolution can be improved in a broader context using quantum light, e.g., Raman and two-photon absorption spectroscopies.

REPLY: we thank the reviewer for suggesting these interesting references, which now have been added to the revised version of the manuscript.

We added a new sentence in the introduction: “Pathway selectivity [17], time-energy resolution [18] and molecular selectivity [19] can also be improved in a broader context using quantum light.”

Reviewer #2

This manuscript presents the first experimental realization of time-resolved fluorescence spectroscopy with entangled photon pairs. Although several theoretical works have emphasized the advantages of entangled light for time-resolved spectroscopy, an experiment had not previously been carried out—primarily because the extremely weak nonlinear signals were expected to make such measurements impractical. By combining fluorescence detection with a TWINS interferometer, the authors now demonstrate that these measurements are indeed feasible. While the spectral information obtained in this measurement does not exceed what can be accessed with state-of-the-art time-resolved laser spectroscopy, the study is nonetheless highly significant: it lifts time-resolved spectroscopy with entangled photons from theoretical proposal to proven experimental technique. Provided the comments below are addressed, I believe the manuscript is well suited for publication in Nature Communications.

We thank the reviewer for the accurate summary of our work, for recognizing our study to be “highly significant” and for judging that “it lifts time-resolved spectroscopy with entangled photons from theoretical proposal to proven experimental technique.” In the following, we address all the technical comments by the reviewer.

[Comment 1]

Please add a brief explanation of why the PBS placed next to the objective in Fig. 1(a) is used instead of a dichroic mirror. I assume the PBS ensures that horizontally polarized light enters the subsequent TWINS interferometer for further polarization control, but researchers performing experiments with microscopes would normally employ a dichroic mirror at this position. A short clarification would therefore make the setup easier to understand.

REPLY: The PBS is indeed there so that we can easily separate the signal and fluorescence photons and send polarized light to the input of the birefringent interferometer, as required for its operation. Alternatively, a dichroic mirror could also be used. We now explain this in the caption to figure 1.

[Comment 2]

Please plot the raw photon counts in Fig. 5 instead of normalizing the data to unity.

REPLY: We have replotted all figures in Fig.5 that now show the actual raw photon counts (coincidences)

[Comment 3]

Several key theoretical papers on time-resolved spectroscopy with entangled photons are missing from the Introduction:

[A] A. Ishizaki, J. Chem. Phys. 153, 051102 (2020).

[B] L. Ko, R. L. Cook & K. B. Whaley, J. Phys. Chem. Lett. 14, 8050 (2023)

[C] J. J. Fan, Z. Y. Ou & Z. Zhang, Light Sci. Appl. 13, 163 (2024)

[D] Y. Fujihashi, O. Iso, R. Shimizu & A. Ishizaki, arXiv:2502.02073 (2025)

In particular, while reference [D] treats SPDC driven by a pulsed laser rather than by a continuous-wave source, both studies are conceptually similar in that they exploit entangled photon pairs to obtain time-resolved fluorescence spectra. The authors should therefore comment this connection in the manuscript.

REPLY: we thank the reviewer for pointing out these interesting and relevant references. They have now been added to the manuscript together with other references suggested by the other reviewers. We have added a new paragraph to the introduction:

“Pathway selectivity [17], time-energy resolution [18], molecular selectivity [19] and two-dimensional spectroscopy [20, 21] can also be improved in a broader context using quantum light. Theoretical studies have also shown that, in principle, careful choice and shaping of classical probe states can reproduce some aspects of quantum measurements performed with classical pump pulses and single photon Fock-state probe states [22].”.

Reviewer #3

Alvarez-Mendoza and co-workers describe an experimental technique exploiting pairs of entangled photons generated by a CW laser source to realize time and frequency resolved TCSPC measurements. The technique is applied to two example applications, claiming an advantage in the signal-to-noise versus typical integration times for similar measurements demonstrated in the past. The combination of reasonably short integration times with the use of a CW illumination scheme is appealing in terms of overall cost and simplicity of the experimental apparatus. Frequency resolved measurements are also presented, even though performed at the cost of a linear increase of the measurement time. The manuscript is a timely contribution dealing with an interesting topic. The idea for this work is somehow indebted to a previous paper which confirmed that energy transfer in photosynthetic antenna complexes works on a single-photon basis. This work is correctly acknowledged in the manuscript, with a clear indication of how the current work advances over the previous one.

We thank the reviewer for their accurate summary of our work, for judging it “appealing” and “a timely contribution dealing with an interesting topic”. In the following, we address all the technical comments by the reviewer.

On the other hand, the novelty and technological advantage of the current implementation over previously demonstrated ones [31-33] is not clearly discussed nor quantified. With respect to previous works, this manuscript adds frequency resolution, however at the cost of repeating several measurements to reconstruct spectral information. This may be (trivially?) possible also with previously shown implementations, suggesting the lack of an inherent advantage in the proposed measurement scheme. For publication on Nature Communications, it is important to show that the newly proposed method is not incremental with respect to previous results.

REPLY: To answer the reviewer’s request on the novelty with respect to previous implementations, we show Table T1 below that outlines the main findings and parameters reported in Refs 31-33 and makes a direct comparison with our work.

There are two points to be underlined, one technical and one more conceptual.

At a technical level, we show improvements in acquisition times and temporal resolution. We argue that this improvement is not incremental when referring to increases in resolution of $>10\times$ in lifetime and $>10\times$ - $100\times$ in acquisition time. We underline that e.g. in Ref. 33, the authors conclude that integration times of sub-1minute will require 10^9 photons/second detection rates and then move on to conclude that this is therefore not possible with standard EPP sources and single-photon detectors. Our fluorescence lifetime measurements recorded with sub-second acquisition time clearly demonstrate that **we achieved what was thought to not be possible based on previous literature**. These improvements do also then actually represent a step change in terms of capability and transform the use of SPDC for lifetime sensing from a “curiosity” to an actually relevant experimental tool. It is also this improvement that allows us for the first time to simultaneously measure the lifetime and emission spectrum (i.e. spectrally resolved lifetime). This is the reason why we refer to “**spectroscopy**” in the title of our manuscript. This is not merely “lifetime” or “FLIM” but it is actual “time-resolved **spectroscopy**”. Without the reported improvements, adding the new axis of spectrum would not have been possible.

At a slightly more conceptual level, we underline that all previous works relied only on temporal correlations and indeed Ref.32 for example expands on how one might achieve similar results with purely classical correlations that, in principle, could be sufficient to reproduce their results. Their conclusion is that one could not achieve nanosecond resolution with current options for generating classical correlations. In our case, beyond their observation on the feasibility of using classical correlations, we do take things one step further and harness simultaneous temporal and frequency correlations – the herald photon is filtered spectrally so as to ensure that we collect fluorescence only from events triggered by pump (signal) photons with the desired frequency. We are therefore conditioning our fluorescence photon measurements simultaneously in time and frequency (of the pump photon). This is a stronger use of the quantum correlation properties of the SPDC photons. We also echo a comment on Ref.32, i.e. with these methods the aim is not to find an aspect of the quantum entangled photons that can only be explained in the context of quantum theory, but rather to find unique applications that capitalize on the advantageous characteristics of these sources. In this context, our results take this objective a significant step further.

PAPER REF:	IRF	resolution/accuracy	precision	SNR	acquisition time	molecule	spectroscopy	correlations
31 - Entangled Photon Correlations Allow a Continuous-Wave Laser	350ps	measured 1ns lifetimes	consistent results obtained only for >100 second integration. Std > 100 ps below 10 seconds. Lifetime varies by more than 2x for <10second integration. 1hour integration needed for 10 ps std	of order 0.05 for 20 minutes acquisition	15minutes quoted as minimum acceptable	IR-140 (QY=0.17)	no	time only
32- Fluorescence lifetime measurements using photon pair correlations generated via spontaneous parametric down conversion	of order 300 ps	measured 4 ns lifetimes. Authors say that short lifetimes are possible but do not show this. 200 ps std on fits at lower concentrations	authors only quote >100 photon pairs/second	hard to estimate (missing parameters), possibly of order 0.01 in 10 seconds	presumably 1-10 seconds, not specified	rhodamine 6G (QY=0.95)	no	time only
33- Benchmarking of fluorescence lifetime measurements using time-frequency correlated photons	3.65ns	365ps (estimated as 10% deconv, not actually measured)- 100 ps std at 1 min integration	365 ps estimated as 10% deconvolution resolution (but not actually measured)	of order 0.03 for 1 hour measurement	14 hours, 20 minutes quoted as possible	indocyanine green (QY=0.14)	no	time only
our work	260 ps	10 ps std on measurements with measured lifetimes as low as 100 ps.	10 ps std from lifetime fitting. No systematic error or increase in std for fast sub-second acquisitions - lifetime prediction robust	of 0.05 for 2 second measurement	0.6 seconds with robust results.	LH1 and LH2	yes	frequency and time

Table T1: comparison of the results presented in our paper with those previously achieved with EPPs generated by a CW laser.

Changes to the manuscript: we have added the following point to the discussion/conclusion section:

“We show improvements in acquisition times and temporal resolution. These improvements represent a step change in terms of capability compared to previous work and transform the use of SPDC for lifetime sensing into a relevant experimental tool. It is also this improvement that allows us for the first time to simultaneously measure the lifetime and emission spectrum (i.e. spectrally resolved lifetime). Beyond this, we also take things one step further in terms of exploiting the quantum features of SPDC and harness simultaneous temporal and frequency correlations – the herald photon is filtered so as to ensure that we collect fluorescence only from events triggered by pump (signal) photons with the desired frequency. We are therefore conditioning our fluorescence photon measurements simultaneously in time and frequency (of the pump photon). Whilst the fluorescence cancels the quantum correlations imprinted into the SPDC photons so that we should not expect to observe any quantum features in the final measurement, the goal of this work is to show that the joint temporal and spectral correlations of EPP sources can be used for time-resolved spectroscopic measurements with measurement timescales that are compatible with expectations from existing classical methods. In this context, our results take this objective a significant step further.”

In general, the Authors should clarify:

- how much of the signal-to-noise ratio enhancement is provided by the CW illumination scheme (which is quite recent, but not new), how much by the quality of the detection scheme (which relies on commercial detectors), the NA of the objective, or the exploitation of the quantum nature of the photon pair (besides their temporal/frequency coincidence). In terms of detection, I understand that part of the signal-to-noise improvement ratio comes from the

decision to use SPAD detectors rather than SPAD arrays. On the other hand, using a single SPAD detector pushes the duration of spectral measurements into the hours. How does this compare to the previous literature?

REPLY: The overall gains are a combination of multiple optimizations in the brightness of the EPP source, the high efficiency of the single-photon detectors and the care in choosing the optics. It is hard to identify any precise improvements compared to previous work, as the full details required to assess this are not provided in the papers. So, all we can say is that our measurements very clearly show that it is possible to measure fluorescence lifetimes in regimes that others have indicated would not be possible (see e.g. comment above re. Ref. 32) and then extend this approach also to full spectroscopy. We provide the “recipe” in our manuscript that we followed in building our setup so that others can then follow the same approach. As for the spectroscopy measurements and the use of a SPAD camera versus a single pixel, it is hard to provide a proper direct comparison with a grating spectrometer approach, as the final result depends on many specific details, but we can try to give some estimates under the assumption that we are comparing the two approaches for the same acquisition time. Specifically, we can base the considerations around the SNR:

$$SNR_{pixel} = \frac{S \cdot T}{\sqrt{(S + D_{pixel}) \cdot T}} = \sqrt{T} \cdot \frac{S}{\sqrt{S + D_{pixel}}} \quad \text{Eq.1}$$

where S= signal, D = dark counts, T = measurement time.

Compared to the TWINS-single pixel case, the grating case leads to reduction of S to S/m. The reduction factor $m = N/q$ contains two terms: N is the number of pixels on the camera, q is the (typical) reduction in photon detection probability in the SPAD array compared to a single pixel SPAD. The reduction q arises as a result of the lower fill factor of the active pixel area in SPAD arrays (of order 1%-50%) compared to single pixels (100%) and reduced quantum efficiency in SPAD arrays (max typically around 50% but as low as 1-10% at 800nm) compared to a single pixel (as high as 70%-80%, also at 800nm). We could also account for grating efficiency but if we take this to be ~50% then we can ignore this loss as we have a similar loss also for the TWINS as this requires polarised light at the input.

For a grating coupled to a SPAD camera we have

$$SNR_{camera} = \sqrt{T} \cdot \frac{\frac{S}{m}}{\sqrt{\frac{S}{m} + D_{camera}}} = \sqrt{T} \cdot \frac{S}{\sqrt{S + D_{camera} \cdot m}} \quad \text{Eq.2}$$

Taking the ratio of Eq.1 and 2 then gives us a ratio of the SNRs for the same signal integration time T:

$$\frac{SNR_{pixel}}{SNR_{camera}} = \frac{\sqrt{S + D_{camera} \cdot m}}{\sqrt{S + D_{pixel}}}$$

We note that regardless of the specific numerical values, this ratio is always > 1 .

We also note however, that we must also scan the TWINS spectrometer, typically 100 or 200 scan points are used in our experiments.

If we combine this with the reasonings above we conclude that if we accumulate signal on the camera for the same time allowed to scan the TWINS by M steps, then the ratio R of the SNR for the single-pixel TWINS and grating spectrometer coupled to a SPAD array can be written as:

$$R = \frac{SNR_{pixel}}{SNR_{camera}} = \frac{\sqrt{S + D_{camera} \cdot m}}{\sqrt{M} \sqrt{S + D_{pixel}}}$$

We can provide estimates by inserting values, based on our own measurements in 1 second, see eg Figure 5, with $S \sim 100$ counts. We note that R does not depend on time (thus justifying the choice to work with values estimated for 1 second acquisition) as both the signal and the dark counts grow linearly with time and this dependence cancels out.

For our single SPADs we have $D_{pixel} \sim 50$ counts/second and 70% photon detection probability (Excelitas SPSM-NIR) and if we take a best-of-market linear SPAD array so as to maximise fill factor (Pi- Imaging SPAD lambda) we have 320 pixels with $D_{camera} \sim 250$ counts/second and $q=0.08$ at 800nm (80% fill factor, 10% QE), i.e. $m=4000$, we then have $SNR_{pixel}/SNR_{camera} = 8$. As a further example, one might instead consider another state-of-the-art camera, e.g. SPAD512 (PI Imaging), 512 pixels, $D_{camera} \sim 25$ counts/second, $q=0.05$ at 800nm (50% fill factor, 10% QE), i.e. $m=10240$ at 800nm: $SNR_{pixel}/SNR_{camera} = 4$. These estimates based on actual values of available technology indicate that for the same total acquisition time, the TWINS (coupled to a single SPAD pixel) SNR outperforms a grating spectrometer with a SPAD array.

Regarding the actual total acquisition time in our work for the full time-resolved spectroscopy measurements: the spectrum shown in Fig.4b was taken under the same acquisition conditions shown in Fig.5a, i.e. 1 minute for each point. We acquired 200 scan points, which equates to 200 minutes=3.3 hours. This measurement was carried out under these conditions simply to showcase the best possible outcome. However, as can be seen in Fig.5, good information can be obtained in 2 seconds and if acquiring over 100 points, we can obtain a full time-resolved spectrum in 3.3 minutes. Rather than leave this as a speculative point or extrapolate this measurement from previous data, we actually **took new data** with these parameters to properly experimentally verify that we can take a full time-resolved spectrum of LH2 in 3.3 minutes. This new data is shown in the insets to Figures 4b and c. This shows that our approach can be extremely effective.

Below we show as Figure R4 an additional data set taken with slightly longer acquisition, i.e. 5 seconds per scan point, leading to a total acquisition time of 8 minutes 20 seconds.

Figure R4: time- and frequency-resolved fluorescence from LH2 obtained upon single-photon excitation and a measurement time of 8 minutes and 20 seconds.

We have added a Methods section that summarises the derivations shown above and the calculations.

We have also added a new paragraph to the manuscript:

“The choice of a Fourier Transform Spectrometer approach is based on SNR considerations alone. With the TWINS, we collect all the light with a single SPAD that is optimized for photon collection (100%) and quantum efficiency (70% at 800 nm). This ensures that high SNR time traces at each interferometer position can be acquired for short times with high signal and low dark counts. The compromise is that acquisition times are of course lengthened by the requirement to scan over interferometer delays in our experiments. A different approach could be to use a grating spectrometer coupled e.g. to a SPAD array. In the latter case, quantum efficiencies are typically a factor 10x lower and fill factors can typically also be a factor 10x lower compared to a single pixel SPAD. This brings a factor 100x penalty in SNR. Moreover, the spectrum is spatially dispersed resulting in an N-fold signal reduction, where N is the number of pixels or spectral points measured. This signal can be recovered by increasing the acquisition time so as to match the total TWINS acquisition time (that is lengthened due to the M scan points)

We can estimate the ratio R of the SNR for the single-pixel TWINS and grating spectrometer coupled to a SPAD array as

$$R = \frac{\sqrt{S + D_{camera} \cdot m}}{\sqrt{M} \sqrt{S + D_{pixel}}}$$

where $S \sim 100$ counts is the measured signal in 1 second, D is the sensor dark counts (for the camera and single pixel), $m = N/q$ and q is the photon detection probability reduction in the camera compared to a single pixel. Inserting values from our experiments for the TWINS case and using published values for state-of-the-art SPAD cameras (see Methods), we find that for the same total acquisition time, $R \sim 4-8$, indicating that there is an advantage in using a Fourier Transform approach over the competing solution using a grating spectrometer coupled to a SPAD camera.”

- If the observed advantage comes mainly from the detection scheme, I wonder whether it could be translated to other "classical" techniques. Pump-probe and optical-gating measurements can also work with very low probe intensities (despite requiring still pulses operation).

REPLY: We do not think that our approach, based also on the discussion above, would directly transfer to the use of classical sources. The main obstacle would be the timing information which would not be possible to retrieve with a CW classical laser alone (i.e. without SPDC and exploitation of temporal correlations between signal and idler photons). This can only be recovered using pulsed lasers. Of course one could exploit quantum correlations between signal and idler photons to bring the sensitivity of pump-probe spectroscopy below the shot noise limit, but that would be a completely different experiment.

- Spectral resolution is provided by the TWINS device, which however introduces also insertion losses and consequently a trade-off on the temporal resolution. Can the Authors comment on the possibility, advantages and disadvantages of using instead dispersive elements, spectral filters, etc. as alternative ways of achieving spectral information? Would there be any specific penalty, considering that the spectral information is still reconstructed based on several (how many?) sequential measurements?

REPLY: This is a good point and we have replied to this point in detail above and in the changes made to the main text and methods section. It certainly seems to us that there are several advantages in using a Fourier Transform spectrometer approach, specifically in SNR, detector efficiency and total acquisition time (as detailed above). In the revised version of the manuscript, in fact, we report a full time- and frequency-resolved spectroscopy measurement acquired in just 3.3 minutes. We have added a comment in the manuscript, described above in reply to the comment about the use of SPAD arrays versus single pixels.

- In their introduction, the Authors present some possible strategies for a quantum advantage, such as the possibility of overcoming "granular (shot) noise" or to "reach the so-called Heisenberg limit". However, it is not clear whether either of these two quantum-specific enhancements are demonstrated in this work, or whether only the temporal/frequency correlation is used.

REPLY: We apologise for any confusion – we only mentioned the Heisenberg limit in the introduction as examples of where quantum properties have played a role in the past in imaging. This was not meant to say that reaching the Heisenberg limit was also achieved in this work. Here we use time/frequency correlations and, importantly, any quantum correlation properties are not transferred over to the fluorescence photons. In order to avoid confusing future readers, we have removed the sentence from the introduction where we mention the Heisenberg limit.

- The comparison against the results already presented in references [31-33] should be made more quantitative. The two aspects of signal-to-noise ratio vs integration time, and that of temporal resolution should be discussed separately. Regarding signal-to-noise ratio and the uncertainty in the determination of decay lifetimes reaches a level of 0.01 ns within about ~100

s in these previous papers. The Authors claim similar uncertainty levels with shorter integration times between 30-50 s, leading to a possible improvement of a factor 3-4 with respect to these previous reports. However, it is unclear what the uncertainty of 0.01 ns actually represents in the measurements shown by the Authors, since other descriptors of the fitting goodness are missing. Such a low uncertainty seems also not compatible with the IRF deconvolution step, coarseness and noise level of the curves shown. This is a critical factor as the careful determination of this uncertainty underpins the main claim of this work. Previous works provided a more quantitative overview on these aspects, rather than just showing qualitatively the noise level at different integration times.

REPLY: This is a good point. It is not easy to provide exact estimates for all of the parameters, specifically resolution (or accuracy), precision (of the lifetime values) and SNR (of the intensity values). The table below is an attempt to do this to the best of our ability in extracting the data values from the references. We believe that the overall comparison supports our work very favorably.

The uncertainties reported in our results (similarly to those reported in Ref.31 and 33) are the standard deviations from the deconvolutions. Our IRF is 260ps. We are not certain why this is considered to not be compatible with deconvolution. One often finds quoted that deconvolution can only recover lifetimes that are a factor 10x shorter than the IRF, but this depends critically on SNR and can be significantly better or worse than that. Moreover, that is a constraint on the accuracy (i.e. shorter lifetimes might be affected by a systematic error, such as that seen e.g, in Ref. 33 when SNR is decreased). The std errors we are referring to here are related to the precision.

To clarify the meaning of the errors reported in our work, we have added a sentence:

“The uncertainty reported in the lifetime values refers to the std of the fits and relates to the precision with which the lifetime is measured.”

We have also added the SNR values in each graph in figure 5 and a comment in this in the main text.

PAPER REF:	IRF	resolution/accuracy	precision	SNR	acquisition time	molecule	spectroscopy	correlations
31 - Entangled Photon Correlations Allow a Continuous-Wave Laser	350ps	measured 1ns lifetimes	consistent results obtained only for >100 second integration. Std > 100 ps below 10 seconds. Lifetime varies by more than 2x for <10second integration. 1hour integration needed for 10 ps std	of order 0.05 for 20 minutes acquisition	15minutes quoted as minimum acceptable	IR-140 (QY=0.17)	no	time only
32- Fluorescence lifetime measurements using photon pair correlations generated via spontaneous parametric down conversion	of order 300 ps	measured 4 ns lifetimes. Authors say that short lifetimes are possible but do not show this. 200 ps std on fits at lower concentrations	authors only quote >100 photon pairs/second	hard to estimate (missing parameters), possibly of order 0.01 in 10 seconds	presumably 1-10 seconds, not specified	rhodamine 6G (QY=0.95)	no	time only
33- Benchmarking of fluorescence lifetime measurements using time-frequency correlated photons	3.65ns	365ps (estimated as 10% deconv, not actually measured)- 100 ps std at 1 min integration	365 ps estimated as 10% deconvolution resolution (but not actually measured)	of order 0.03 for 1 hour measurement	14 hours, 20 minutes quoted as possible	indocyanine green (QY=0.14)	no	time only
our work	260 ps	10 ps std on measurements with measured lifetimes as low as 100 ps.	10 ps std from lifetime fitting. No systematic error or increase in std for fast sub-second acquisitions - lifetime prediction robust	of 0.05 for 2 second measurement	0.6 seconds with robust results.	LH1 and LH2	yes	frequency and time

Table T1: comparison of the results presented in our paper with those previously achieved with EPPs generated by a CW laser.

- Similarly, in terms of resolution limits, some of the previous experimental works cited by the Authors claimed to be theoretically able to measure decay times shorter than those measured experimentally in this work. Hence, the key claim that previous reports showed "significantly lower temporal resolution and longer measurement time" does not appear to be sufficiently substantiated.

REPLY: We refer to the table provided above, which summarizes the actual values reported in the cited papers. It is interesting to note that e.g. Ref.33 essentially concludes that the measurements (specifically, the SNR for our measurement times) we report should not be possible. This is just to highlight that making theoretical predictions of what could be achieved should always be taken with a grain of salt. Said differently, we feel that predicting that one could maybe make a measurement with a certain accuracy is not the same as actually doing it. In this work, we report only on experimentally achieved results that outperform previously reported experimental results by factors 10-100x.

- In the comparison against the results in [31-33], another important point which is not commented upon in this work is that of the dye concentration. Signal-to-noise and integration time depend critically on the molar concentration, and integration times should be further normalized by this quantity. These aspect should also be taken into account when comparing the results to those in [31-33].

REPLY: We agree with the reviewer. We report the concentration values in the Methods section and specifically for the dyes, we also have added a comment to underline that in one experiment we use the same dye as Ref. 33 (now, Ref. 37) with very similar concentrations (1.5 versus 1.3 mMol/L).

- Finally, I have a couple of comments on the current wording of the abstract, which is potentially misleading. A statement such as "we overcome this paradigm by exploiting quantum correlations to perform time-resolved spectroscopy with entangled photons" seems to claim a

level of novelty that was already demonstrated in previous works (perhaps the sentence can be removed altogether without altering too much the sense of the abstract).

REPLY: We agree with the reviewer and, in response to other comments, we have indeed removed this sentence/

Similarly, the abstract mentions twice the fact that "experiments are performed under conditions comparable to real-world sunlight illumination". Since solar light is not entangled, this statement refers solely to average intensity levels (impinging on the non-linear crystal? or interacting with the sample?), or would this be a hint at the change of generating entangled photon pairs using solar light on the crystal

REPLY: This sentence is indeed the subject of other comments and has been revised. And now reads

“experiments are performed under illumination intensity conditions comparable to real-world sunlight illumination”.

- "t = 56°" should be "T = 56°C"

REPLY: corrected.

Pending suitable revisions to clarify these points, the paper can be reconsidered for publication at Nature Communications.

We thank the reviewer for the very useful comments, which have greatly helped us to improve the clarity of our presentation, and which we believe we have fully addressed in the revised version of the manuscript.

Reviewer #4

...In my opinion, however, entanglement plays no role in this experiment. Correlations between the timings of the signal and idler photons certainly do play a role, but this is not adequate to say that the experiment “exploits entanglement”. Entanglement is a coherent effect that requires measurement basis changes to be made in both fields to manifest, but here the idler photon is only ever subjected to direct time-resolved detection.

REPLY: We understand the point being made here but only partially agree.

We certainly agree with the referee that the unique quantumness of entanglement emerges from two points, i.e. measurements across (at least) two variables and made for various settings.

When combining all these measurements, one then realizes e.g. that the resulting distributions or measurement results cannot be explained classically. And indeed, we acknowledge the point that we have not performed something like a Bell inequality measurement where we might measure the spectrum and arrival time across various choices of frequency and time bins.

However, we are leveraging the simultaneous correlations across both frequency and time. This is the main reason why we refer to “entanglement” in our work – we simultaneously herald both in time and frequency (not just time).

We acknowledge that during the measurement, the time basis measurement is performed for multiple time bins whereas the frequency basis is fixed. Nevertheless, it is very hard to imagine a classical setup where one uses a CW laser and obtains both temporally and frequency correlated

photons with the characteristics required here. We underline the constraint of the CW laser in our measurements.

Our point therefore is that we are exploiting the correlation properties that arise from the fact that the photons, at the point of generation, were entangled. As far as we are aware, this is routinely summarised in literature by saying that we “exploit entanglement”. **Our technique does not require a measurement of the degree of entanglement (or e.g. a Heisenberg uncertainty violation) in the way e.g. that cryptography might. But we do need the ω -t correlations and non-separability that come from entanglement in order to do the measurements (with a CW laser).**

As a side note, the debate as whether this might be possible with a classical laser source is not that relevant in the sense that, in this work, we are not exploiting classical properties but the properties that arise as a result of the entanglement. That is our claim. All that said, we are not aware of any classical route by which it would be possible to replicate these exact measurements with a CW pump laser.

We have added the following text to the manuscript to clarify this important point raised by the reviewer:

“We underline that the measurements shown here with a CW pump laser exploit the simultaneous correlations in time and frequency that are a direct consequence of the entanglement of the photons at the point of generation in the nonlinear crystal.”

The function of the idler field, as far as this experiment goes, is therefore simply to convey classical timing information about when the single-photon signal pulses are emitted.

Most of the results of this experiment, in my view, could be replicated with ultrafast classical pulses with the same bandwidth as the emission spectrum of the SPDC source.

REPLY: We agree with the reviewer that time-resolved fluorescence measurements can be performed with a pulsed laser. Indeed, this is how they have performed traditionally since decades. The point of our work though is that we perform the measurements with a CW laser, and this would not be possible with a classical system. There is no classical scheme that we are aware of that would allow to perform time-resolved measurements with a CW laser and even less so when we also include the fact that we are also simultaneously exploiting correlations in frequency and time by heralding arrival time and spectral properties of the pump photon. To further clarify our point, we have added a sentence

“Whilst traditional time-resolved measurements are performed using ultrafast pulsed laser systems, here we demonstrate that by exploiting the frequency and time correlation properties of entangled photons, it is possible to perform high SNR measurements in short acquisition times using a CW laser.”

The exception is the $g(2)$ measurement of the heralded signal photon, but this does not impact on the other results of the paper, which are not in any way dependent on the single-photon character of the signal field.

REPLY: As explained above, we argue that the results, obtained with a CW laser, are indeed very much the result of the joint (bi)photon nature of the signal and idler fields. Without this, we do not see how one could achieve the same-time resolved measurements shown in our work.

The authors also perform a spectrally resolving measurement on the signal field using a Fourier-transform spectroscopy, implemented by a common-path polarization interferometer inserted into the path of the fluorescence. By splitting the pulse into two copies separated by a variable delay, some timing information is lost (in the sense that more measurements are required to estimate the arrival time), with the payoff that the transmission probability of the photon carries some information about the spectrum. However, many copies are required to obtain a full interference pattern needed to recover the spectrum. FTS is already very widely used so I don't think this an especially novel inclusion.

REPLY: We agree with the referee that FTS is in itself not novel and indeed, the exact TWINS implementation used here is a commercial device. Our point though was to demonstrate that the efficiency of the whole setup is such that it allows spectrally resolved measurements to be possible. This is therefore not a conceptual innovation but a technical innovation. We appreciate that it might not be obvious, but this is not a minor point for any experimentalists working in the field and indeed, it had never been shown before. Specifically, there are two technical points here: 1) entangled photon pairs can be used to perform time- and frequency- resolved measurements of fluorescence; 2) this is best achieved with an FTS rather than with the other option, which would be a standard grating-based spectrometer. The latter requires spatially dispersing the spectral information leading to very low counts and low SNR at each pixel of e.g. a SPAD camera whilst the FTS approach collects all the photons on a single pixel (bucket detector) without spatial dispersion.

We have now added a new section in the Methods and also a paragraph in the main text to explain that, although FTS is not novel in itself, it was chosen as it brings important advantages in terms of SNR.

“The choice of a Fourier Transform Spectrometer approach is based on SNR considerations alone. With the TWINS, we collect all the light with a single SPAD that is optimized for photon collection (100%) and quantum efficiency (70% at 800 nm). This ensures high SNR time traces at each interferometer position can be acquired for short times with high signal and low dark counts. The compromise is that acquisition times are of course lengthened by the requirement to scan over interferometer delays in our experiments. A different approach could be to use a grating spectrometer coupled e.g. to a SPAD array. In the latter case, quantum efficiencies are typically a factor 10x lower and fill factors can typically also be a factor 10x lower compared to a single pixel SPAD. This brings a factor 100x penalty in SNR. Moreover, the spectrum is spatially dispersed resulting in an N-fold signal reduction, where N is the number of pixels or spectral points measured. This signal can be recovered by increasing the acquisition time so as to match the total TWINS acquisition time (that is lengthened due to the M scan points) We can estimate the ratio R of the SNR for the single-pixel TWINS and grating spectrometer coupled to a SPAD array as

$$R = \frac{\sqrt{S + D_{camera}} \cdot m}{\sqrt{M} \sqrt{S + D_{pixel}}}$$

where $S \sim 100$ counts is the measured signal in 1 second, D is the sensor dark counts (for the camera and single pixel), $m = N/q$ and q is the photon detection probability reduction in the camera compared to a single pixel. Inserting values from our experiments for the TWINS case and using published values for state-of-the-art SPAD cameras (see Methods), we find that for the same total acquisition time, $R \sim 4-8$, indicating that there is an advantage in using a Fourier

Transform approach over the competing solution using a grating spectrometer coupled to a SPAD camera.”

It is also not mentioned that the TWINS interferometer also introduces on average 50% loss at the polarizer.

REPLY: This is correct. We have added a comment about 50% loss.

“We note that the TWINS spectrometer requires polarized light at the input and that the polarizing beamsplitter used to collect the fluorescence leads to 50% loss.”

A few other related criticisms:

The authors say: “We can nonlocally control the spectrum of the excitation photons that lead to detected fluorescent photons by placing on the idler an interference filter with 10-nm bandwidth at 860 nm.” It is inaccurate to describe this as any kind of “nonlocal control” as the marginal properties of the signal field are unchanged by locally filtering the idler. Rather, by choosing which photons can trigger heralding events, simply a different subset of signal photons give rise to coincidences.

REPLY: We agree with the referee. We are not “controlling” anything nonlocally. We have modified the text in the manuscript which now reads

“By placing a spectral filter, e.g. centered at 860 nm, on the herald photon detector, we can selectively measure only the fluorescence photons that were excited by signal photons that have a conjugate wavelength to the herald photon, i.e. 800 nm in this case”.

I feel that some of the introduction is not relevant. The authors emphasise in the first line of the abstract that most experiments in ultrafast spectroscopy are carried out with much higher intensities than are found in nature, before claiming to “overcome this paradigm by exploiting quantum correlations”. They elaborate in the introduction that “[r]educing the excitation intensity to more realistic values would result in vanishing nonlinear signals for classical time-resolved experiments,” which is true for nonlinear/multi-photon interactions. This is not relevant to the present paper, though, which is investigating a single-photon excitation process that is linear in the intensity. In fact, the key result of Li et al is that for precisely this process, the fluorescence signal from truly single-photon excitation is not only non-vanishing, but measurable; and indeed many other experiments have carried out spectroscopy at the single-photon level.

REPLY: We were originally only trying to give some background and provide further justification as to why one might want to use to entangled photons to pump fluorescence. That said, we fully agree with the reviewer’s comments and in general, that the phrasing was misleading. We have therefore rephrased the abstract and the introduction accordingly – these now focus on the main claims of the work, i.e. the demonstration of efficient, i.e. fast lifetime and spectroscopic measurements using SPDC photons.

REPLIES TO REVIEWERS.

We thank the reviewers for taking the time and effort to look at our manuscript and provide feedback. We have copied below their comments together with our replies and list of changes to the manuscript.

Reviewer #1 (Remarks to the Author):

The authors have addressed all my comments, and I am satisfied with most of the answers. I think there are still two questions that have to be clarified.

1. In Fig.1a as added, it shows that the spectrometer/grating is involved in i-photon detector, to implement the frequency resolution. This seems confusing to me--I would assume the spectrometer is equipped within the s-photon detector, whereas the i-photon detection offers a temporal gate within a narrow time window. So which detector includes the spectrometer? The authors should clarify.

REPLY: The TWINS spectrometer is placed in the arm that is indicated as “signal (800 nm)” and is indeed as the reviewer would expect. The idler photon (also indicated in the figure and labelled as “idler (860 nm)”) does not have a spectrometer (only an interference filter that is used for heralding the spectrum of the signal photon, as explained in the text), also as the reviewer expected. This signal interferometer was indicated in Fig.1b but is now also added in Fig.1a. We have also updated the figure caption – we think this should clarify the reviewer’s doubts.

2. The authors should better make an estimation of time-frequency scale achieved in present work, i.e., in terms of $\Delta\omega * \Delta t$.

REPLY: the $\Delta\omega * \Delta t$ of the photons is simply given by the SPDC process which generates photon pairs with a mutual coherence time that is given by the inverse of the phase matching bandwidth of the process. This is best seen e.g., in HOM interference where the biphoton interference peak is typically given simply by the inverse of the bandwidth. However, our actual measurements are performed using TCSPC electronics (rather than quantum interference between the photons), so all time-responses are convolved with the electronic response times of the detectors. Therefore, we see e.g. that although SPDC will generate photons with correlation times of the order of 100 fs (corresponding to the 15-20 nm bandwidths), the actual IRF measurements that include electronic response times are of order 100-300 ps, i.e. a factor $\sim 1000x$ longer than the actual expected photon correlation times. We have added a comment to this effect in the caption to figure 2 where we show the measured IRFs and spectra. The comment reads: “The correlation times measured by TCSPC are limited by the IRF of the detectors and are significantly longer than the intrinsic bi-photon correlation times that arise from SPDC.”

Reviewer #2 (Remarks to the Author):

The authors have thoughtfully addressed my questions. I believe the paper is suitable for publication.

Reviewer #3 (Remarks to the Author):

The Authors have comprehensively addressed all the points raised, which resulted in a significant improvement of the manuscript clarity and relevance. I therefore recommend the manuscript for

publication without further amendments.

Reviewer #4 (Remarks to the Author):

The authors address some of my points, and I am happy to see they have revised some of their claims in response to my comments. My fundamental criticism remains however: the paper pitches itself as a demonstration of a new type of measurement where quantum entanglement plays a fundamental role, which I do not agree is the case.

I do not agree with the authors' claim that just using the spectral or temporal correlations individually from an SPDC source are 'routinely summarised in literature by saying that we "exploit entanglement"' - at least, not correctly. The considerable interest in entanglement-enhanced spectroscopy is driven by the promise of a coherent boost in sensitivity that is predicted from entangled photon pairs, such as in multi-photon spectroscopy. Nothing like this is demonstrated here. The timing precision here is obtained by the temporal correlations in the photon pairs emitted by parametric fluorescence, but there are many classical sources that also exhibit temporal correlations, and pulsed sources can be trivially correlated with an electronic gating signal.

REPLY: we agree with the reviewer that most certainly time-resolved spectroscopy can be performed with classical, pulsed laser sources. We are of course not intending to claim that time-resolved spectroscopy cannot be achieved by any classical means. Our claim is that we have demonstrated that time-resolved spectroscopy is possible with correlated photon pairs generated by SPDC, that the measurements are now comparable in terms of SNR and acquisition time to classical measurements (thus improving over previous attempts by orders of magnitude) and that this can also be achieved with a CW laser. This last aspect cannot be achieved classically.

The correlations present in SPDC are in a sense an artifact of the quantum nature of the source, but the underlying entanglement itself is not exploited here in any meaningful sense.

REPLY: this is true, we agree.

In response to mine and Reviewer 3's comments the authors have somewhat clarified this with the text added towards the end of p1. But even after these revisions, there is still a strong risk that readers might misinterpret this work as claiming to demonstrate the coherence-boosted proposals that are the real subject of the current interest in quantum-enhanced spectroscopy. Unfortunately, this misunderstanding is already evident in some of the other referee comments.

REPLY: we see the reviewer's point and hoped that previous changes would have dispelled any confusion. To address the reviewer's concern, we have now purged the words 'entangled' and 'entanglement' throughout the text and replaced them with 'correlated' or 'correlations', including in the title, which now reads: "Correlated-Photon Time- and frequency-resolved optical spectroscopy". We have also now added a very explicit discussion in the introduction and in the final paragraph of the manuscript that clarifies that we do not exploit quantum interference nor do we achieve a quantum advantage in traditional sense. We have written in the introduction: "we also underline that this approach does not explicitly rely on entanglement to achieve any form of 'quantum advantage' in the strictest sense (i.e. a measurement that could not be achieved by any conceivable classical means)". And in the conclusions: "In summary, we are not exploiting entanglement directly in the sense of violating classical sensitivity or resolution bounds that might arise from entanglement nor does our approach demonstrate a direct impact of quantum properties on the resolution or SNR. However, we do show that correlations from entangled photons can be used for efficient and fast detection of fluorescence lifetime signals with a CW laser source. This

cannot be achieved by classical means. The sub-second acquisition times using SPDC photon pairs also lay the ground for future work and development of systems that do exploit quantum properties, such as HOM interference for picosecond or sub-picosecond lifetimes but now starting from experiments that can be carried out at rates or acquisition times that are of interest to biologists". We clarify that the correlations allow us to use a CW laser, that could not be achieved classically, but that beyond that, the merit of the work is to demonstrate efficient and high SNR time- and frequency-resolved fluorescence spectroscopy using SPDC. We feel that in doing so, we have hopefully explained in sufficient detail what the work is and what is *not* about.

In their responses the authors have also emphasised the spectral correlations in the fluorescence, but these are not especially important to the measurement. Spectral filtering of the idler and postselecting on coincidences is used to reduce noise from other wavelengths, but this is essentially equivalent to filtering the signal itself. Excessively narrow spectral filtering, in fact, will destroy the temporal correlations that are the basis of the paper due to the Fourier conjugacy of the spectral and temporal bandwidths. A true entanglement-exploiting measurement would necessarily involve a joint measurement on the signal and idler biphoton, such as Hong-Ou-Mandel interferometry. The authors have pivoted to claiming that the advantage of the paper is in the technical expediency of using a CW scheme, harnessing the fact that even with a CW pump, parametric fluorescence contains both tight temporal and broadband spectral correlations. I note that in the original manuscript this was just a passing remark, but has assumed a much greater significance after their revisions. They accept that the use of a pulsed probe and temporal gating can reproduce the timing resolution, and hence the impact of the paper rests on the relatively greater availability and affordability of CW lasers and SPDC sources compared to pulsed lasers.

To end on a positive, I think this is a reasonable point. Indeed, there have been numerous other works in e.g. ghost imaging that do not claim quantum advantage per se, but take advantage of these natural correlations in parametric fluorescence as an experimental tool. A CW laser and nonlinear crystal is certainly cheaper and more convenient than a tunable picosecond laser, which makes this still useful without the questionable claims to be exploiting entanglement. For this reason I think there is some merit in this paper, but it still needs major revisions to dial down the claims to quantum advantage before it could be considered for publication. Even then, the idea is essentially methodological and I would be reluctant to recommend it for publication in such a high impact journal.

REPLY: we hope the reviewer will agree that the current version leaves no room for misinterpretation. We do however disagree that demonstrating such a significant improvement over previous attempts to harness correlated photons for lifetime measurements is marginal. Previous experimental was 'patchy' to say the least, with typically very poor SNR and very long acquisition times, even when using pulsed laser sources. We were also somewhat surprised to see how many theoretical papers have been published in the last few years alone, claiming benefits and various opportunities for the use of SPDC photons for lifetime and time-resolved spectroscopy – 15 theory papers versus only 4 papers attempting experiments. We believe this indicates the potential for an emerging application area for entangled photons that, in our opinion, was precisely missing the input from our work to demonstrate that these measurements, and hence also the various theoretical proposals, are indeed actually possible. We also agree with the reviewer that a joint measurement of the signal-idler biphoton, such as in a HOM interferometer, would be a natural follow-up step to fully exploit entanglement. As we discuss in the conclusions, we believe that our result marks an important milestone in view of such experiments, by demonstrating that time-resolved experiments with SPDC photons can be performed with fast acquisition times of the order of seconds, which are relevant for applications in biology. Taking all this in consideration, we remain convinced that this work warrants publication in Nat.

Comms.

Review report for manuscript NCOMMS-25-33978

The manuscript entitled “Entangled-photon time-and frequency-resolved optical spectroscopy” presents an experimental study of quantum-light spectroscopy for excitation energy transfer in biological complexes. The entangled photon pairs were used for extracting the information about excited-state dynamics of the LH2 complexes, in light of the pump-probe scheme. Using the quantum entanglement and photon-coincidence counting, the signal shows a time-frequency-resolved feature for monitoring the excitation energy transfer in the LH2 photosynthetic antenna, where the time-energy conjugation was lifted. The s photons pump the LH2 to electronic excited states, and the fluorescence was detected in a coincidence with the idler photons which are propagating freely in order to gate the fluorescence temporally. Such quantum spectroscopic tool was demonstrated in two experiments, i.e. EET cascades in photosynthetic membrane and distinguished lifetimes in dyes mixture, revealing a prosperous potential of being an “entangled photon streaking camera”. Although a sub-200 ps temporal scale was achieved, the results in this manuscript are highly important and insightful. I think the experiments performed by authors are a milestone in the journey towards the quantum-light ultrafast spectroscopy. Therefore the manuscript is of a great deal of publishing on Nat. Commun., after a major revision. Some of my comments are listed below.

1. A schematic illustration is needed in Fig.1, in order to show the essence of how the entangled photons pump the molecules and temporally gate the dynamics, with the coincidence counting.
2. I am wondering if the intra-aggregate dynamics within LH2, i.e., between B800 and B850 could be resolved by the signal in Fig.4d, as the horizontal axis is of a scale ~ 1 ns. Since the temporal resolution by the entangled photons is < 200 ps, it would be better to provide the results for a shorter time scale, for example, < 1 ns. This will be very supportive for the conclusion in the manuscript.
3. The scheme in present work is like a “ghost imaging” in time and frequency domains (Well, the ghost imaging renders an elastic scattering of photons, but here the inelastic scattering dominates). The underlying mechanism of the time-frequency resolution in present work is that the fluorescence following the s photon pump has to pass a grating in detector 1 whereas the single-photon pulse in i arm provides the temporal gate; the single-photon detector for collecting i-arm photons does not need to contain the spectrometer. A spectrometer has to be integrated into the single-photon detector collecting the photons from s arm. I am wondering if my understanding is correct? Therefore the present work can go beyond the Nature one in 2023.
4. Have the authors characterized the spectral correlation of the entangled photons, without the samples? The authors should better provide this if yes. The anti-correlation in spectrum (or the correlation in time domain) does play a key role in achieving the time-frequency resolution beyond the Fourier limit.
5. It is worth noting some works related to the present work [Light: Sci & Appls 13, 163 (2024); JCP 160, 104201 (2024); Light: Sci & Appls 11, 274 (2022)]. Pathway selectivity and time-

energy resolution can be improved in a broader context using quantum light, e.g., Raman and two-photon absorption spectroscopies.

Prof. Zhedong Zhang
Assistant Professor
Department of Physics, City University of Hong Kong
Kowloon, Hong Kong SAR